# Directed Graph Generation with Heat Kernels

**Marc T. Law**                                                           *marcl@nvidia.com*
*NVIDIA*

**Karsten Kreis**                                                         *kkreis@nvidia.com*
*NVIDIA*

**Haggai Maron**                                                          *hmaron@nvidia.com*
*NVIDIA*
*Technion*

**Reviewed on OpenReview:** *https://openreview.net/forum?id=60Gi1w6hte*

## Abstract

Existing work on graph generation has, so far, mainly focused on undirected graphs. In this paper we propose a denoising autoencoder-based generative model that exploits the global structure of directed graphs (also called digraphs) via their Laplacian dynamics and enables one-shot generation. Our noising encoder uses closed-form expressions based on the heat equation to corrupt its digraph input with uniform noise. Our decoder reconstructs the corrupted representation by exploiting the global topological information of the graph included in its random walk Laplacian matrix. Our approach generalizes a special class of exponential kernels over discrete structures, called diffusion kernels or heat kernels, to the non-symmetric case via Reproducing Kernel Banach Spaces (RKBS). This connection with heat kernels provides us with a geometrically motivated algorithm related to Gaussian processes and dimensionality reduction techniques such as Laplacian eigenmaps. It also allows us to interpret and exploit the eigenproperties of the Laplacian matrix. We provide an experimental analysis of our approach on different types of synthetic datasets and show that our model is able to generate directed graphs that follow the distribution of the training dataset even if it is multimodal.

## 1 Introduction

The representation of directed graphs (or digraphs) has recently attracted interest from the machine learning community (Clough & Evans, 2017; Sim et al., 2021) as they can naturally describe causal relations (Bombelli et al., 1987), spatiotemporal events using chronological order (Law & Lucas, 2023) or some stochastic processes such as Markov chains (Norris, 1998). Existing work in digraph representation has focused on discriminative tasks such as classification or link prediction. In this work, we consider the task of digraph generation. One possible application is the modeling of new causal systems that follow the same distribution as some given training set.

Most of the machine learning literature on graph generation focuses on undirected graphs (Liao et al., 2019; Niu et al., 2020; You et al., 2018). Their goal is to generate plausible graphs of the same nature as those from some given training dataset (e.g., molecules (Vignac et al., 2023)). Existing approaches can be divided mostly into two categories: *auto-regressive* and *one-shot*. Auto-regressive approaches (Liao et al., 2019; You et al., 2018) start by generating small graphs to which sets of nodes and their corresponding edges are iteratively added until the final graph reaches a certain criterion (e.g., size). On the other hand, *one-shot* approaches generate all the nodes and edges of the generated graphs in a single step. One-shot approaches were shown to be more efficient than auto-regressive ones due to the lack of intermediate steps that can also lead to worse generative performance because of error accumulation at each step and the fact that one-shot

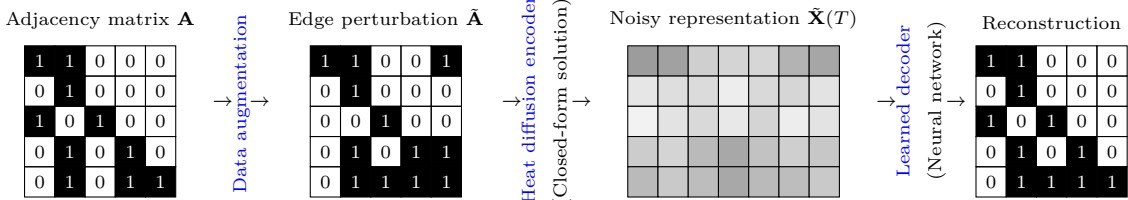

Figure 1: Our framework can be viewed as a **denoising autoencoder**. Our *heat diffusion* encoder maps a perturbed adjacency matrix $\tilde{\mathbf{A}} \in \{0,1\}^{n \times n}$ to a noisy node representation matrix $\tilde{\mathbf{X}}(T) \in [0,1]^{n \times d}$ that is given as input of a decoder that reconstructs the edges. ($n = 5, d = 7$ in the figure)

approaches can directly work with the global structure of the graph via its Laplacian matrix instead of arbitrary subgraphs (Martinkus et al., 2022). Among the one-shot approaches, Spectre (Martinkus et al., 2022) learns the distribution of the most informative eigenvectors of the Lapacian matrices. Nonetheless, Spectre does not generalize to digraphs as it relies on properties of Laplacian matrices that are satisfied only for undirected graphs (see explanation in Section 5). DiGress (Vignac et al., 2023) considers separate representations for nodes and edges to which discrete noise is added. DiGress is formulated as a classification problem such that a denoising decoder classifies the category or existence of edges and nodes. However, DiGress also requires spectral features from Beaini et al. (2021) that are valid only for undirected graphs as they rely on symmetric scalar products. In conclusion, none of the existing one-shot approaches can be easily adapted to digraphs.

**Contributions.** We propose a one-shot approach that generates digraphs in a single step. The training of our framework is illustrated in Fig. 1. Unlike previous approaches, we exploit the eigenproperties of the Laplacian matrix that are valid even when the graph is directed, so we can effectively use its global structure during the generation process. To this end, we propose a denoising autoencoder (Vincent et al., 2008) whose noising encoder is not learned by a neural network but exploits closed-form expressions based on the heat equation for digraphs (Veerman & Lyons, 2020), which effectively encodes the global topological information of the graph into node features. We propose to introduce noise via a nonhomogeneous term that makes our node representations tend to a stochastic matrix with all its elements equal. Our denoising decoder is a neural network trained to reconstruct the original node representations and adjacency matrix of the graph. We explain below the main setup and building blocks of our approach.

## 1.1 Problem definition and overview of our approach

During training, we are given a set of digraphs $\{G_1, \ldots, G_m\}$ represented by their adjacency matrices $\mathbf{A}_1, \ldots, \mathbf{A}_m$. Our goal at inference time is to generate new digraphs that follow the same distribution.

**Training.** At training time, a training graph $G_i$ is represented by its adjacency matrix $\mathbf{A}_i$. First, this matrix is perturbed using data augmentation techniques such as edge perturbation (Ding et al., 2022). The resulting perturbed adjacency matrix $\tilde{\mathbf{A}}^i$ is used as input of our first main component, called *Heat Diffusion Encoder*, which generates some noisy representation, called $\tilde{\mathbf{X}}(T)$. Our encoder produces outputs that are close to a uniform distribution in order to approximate maximum entropy (or lack of information). A detailed description of the heat diffusion encoder can be found in Section 3. Next, our second component, called *Denoising Decoder*, takes $\tilde{\mathbf{X}}(T)$ as input and is trained to predict an adjacency matrix that is as similar to the original (unperturbed) matrix $\mathbf{A}_i$ as possible. The whole training process is illustrated in Fig. 1.

**Sampling new graphs after training.** At inference time, an adjacency matrix is randomly generated and given as input of our autoencoder to generate a novel digraph. We discuss other continuous sampling strategies in Appendix G.2.

## 2  Preliminaries on the Heat Equation

This section introduces our notation and a self-contained introduction to heat diffusion on digraphs. These concepts will be used in Section 3 to define a closed-form expression for our heat diffusion encoder. We refer the reader to Veerman & Lyons (2020) for more details on Laplacian matrices of digraphs, to Chung & Yau (1999); Kondor & Lafferty (2002); Belkin & Niyogi (2003) for heat diffusion kernel-based methods on undirected graphs, and to Appendix C for the details of the equations of this section and the next section. One main difference with other types of data such as images is that graphs lie in a discrete space whose topological information depends on the adjacency of its nodes. Moreover, graphs are often sparse in terms of adjacency. Due to the discrete nature of graphs, calculating similarities between nodes has been a challenge. To tackle this, heat kernel approaches (Chung & Yau, 1999; Kondor & Lafferty, 2002) express the notion of discrete local neighborhood of nodes in terms of a global similarity over the nodes of an undirected graph. More exactly, heat kernels are generalizations of Gaussian kernels where the Euclidean distance that is used in Gaussian kernels is replaced by a more general distance that takes into account the neighborhood information of the Laplacian matrix (Kondor & Lafferty, 2002). The information in node features is propagated along their neighbors via the heat equation. We propose to add a nonhomogeneous term that introduces noise along the heat diffusion process. We explain in Appendix E how our approach generalizes heat kernels to digraphs.

**Notation.** We denote the identity matrix by $\mathbf{I}$, and the all-ones vector by $\mathbf{1}$. We consider a graph $G = (V, E)$ defined by its set of $n$ nodes $V = \{v_i\}_{i=1}^n$ and its set of edges $E \subseteq V \times V$. Its adjacency matrix $\mathbf{A} \in \{0,1\}^{n \times n}$ satisfies $\mathbf{A}_{ij} = 1$ iff $(v_i, v_j) \in E$ and $\mathbf{A}_{ij} = 0$ otherwise. If $G$ is undirected, then we can simply consider that $\mathbf{A}$ is symmetric (i.e., $\forall i, j, \mathbf{A}_{ij} = \mathbf{A}_{ji}$). However, we consider the more general case where $\mathbf{A}$ is not constrained to be symmetric. Although optional, we add self-loops by constraining $\mathbf{A}$ to satisfy $\forall i, \mathbf{A}_{ii} = 1$. The in-degree diagonal matrix $\mathbf{D} \in \mathbb{R}_+^{n \times n}$ is defined so that $\forall i, \mathbf{D}_{ii} = \sum_j \mathbf{A}_{ji}$. In other words, we have $\mathbf{D} := \mathrm{diag}(\mathbf{1}^\top \mathbf{A})$. Let us define the matrix $\mathbf{S} := \mathbf{A}\mathbf{D}^{-1}$. $\mathbf{S}$ is column stochastic (i.e., $\forall i, \sum_j \mathbf{S}_{ji} = 1$ and $\forall i, j, \mathbf{S}_{ij} \geq 0$). We consider in this section that we are given some matrix $\mathbf{N} := \mathbf{X}(0) \in \mathbb{R}^{n \times d}$ where the $i$-th row of $\mathbf{N}$ is the *initial $d$-dimensional feature representation* of $v_i$ (i.e., $\mathbf{X}(t)$ with $t = 0$). The matrix $\mathbf{N}$ could be arbitrarily defined or given. In practice, we train $\mathbf{N}$ jointly with our denoising decoder, and we explain its training process in Section 4 and Appendix B. In this section, we consider that $\mathbf{N}$ is fixed.

**Laplacian dynamics.** We define the negative of the random walk Laplacian matrix as $\mathbf{L} := \mathbf{S} - \mathbf{I} = \mathbf{A}\mathbf{D}^{-1} - \mathbf{I}$. The matrix $\mathbf{L}$ can be viewed as a matrix form of the discrete Laplace operator which approximates the continuous Laplace–Beltrami operator in differential geometry (Belkin & Niyogi, 2003). Given a twice-differentiable real-valued function $f$ defined on some manifold, the Laplace-Beltrami operator is defined as the divergence of the gradient $\mathrm{grad} f$ and provides us with an estimate of how far apart $f$ maps nearby points. Since $\mathbf{L}$ is not symmetric in general, we can use both $\mathbf{L}$ or $\mathbf{L}^\top$ as they have different left and right eigenvectors. We then denote $\boldsymbol{\Delta} \in \{\mathbf{L}, \mathbf{L}^\top\}$. In the main paper, we consider only the case $\boldsymbol{\Delta} = \mathbf{L}$, which is called *diffusion model* in the graph community (Veerman & Lyons, 2020). To avoid confusion with diffusion models in the machine learning literature, we call it *heat diffusion*. We give in Appendix D all the formulae to solve our problem when $\boldsymbol{\Delta} = \mathbf{L}^\top$ (called *consensus model* (DeGroot, 1974)).

**Heat equation.** The global information of the graph $G$ is diffused via the following heat equation:

$$\forall t \geq 0, \frac{\mathrm{d}}{\mathrm{d}t}\mathbf{X}(t) = \boldsymbol{\Delta}\mathbf{X}(t) + \mathbf{Q}(t) \text{ where } \mathbf{X}(t) \text{ is the representation of nodes at time } t \geq 0, \qquad (1)$$

$\mathbf{N} = \mathbf{X}(0)$, and $\mathbf{Q}(t) \in \mathbb{R}^{n \times d}$ is a heat source term that introduces noise in the node representations. If $\forall t \geq 0, \mathbf{Q}(t) = \mathbf{0}$, then $\mathbf{Q}$ is called homogeneous. It is called nonhomogeneous otherwise. The solution of equation 1 can be found in standard textbooks such as Edwards et al. (2020) and is written in equation 2.

For any formulation of $\mathbf{Q}$, equation 1 is solved by the following equation when $\boldsymbol{\Delta}$ is constant over time:

$$\forall t \geq 0, \mathbf{X}(t) = e^{t\boldsymbol{\Delta}}\mathbf{X}(0) + \int_0^t e^{(t-s)\boldsymbol{\Delta}}\mathbf{Q}(s)\mathrm{d}s = \mathbf{Z}(t) + \mathbf{F}(t). \qquad (2)$$

where $e^{t\boldsymbol{\Delta}}$ denotes the matrix exponential of the matrix $t\boldsymbol{\Delta}$. To simplify notation, we define $\mathbf{Z}(t) := e^{t\boldsymbol{\Delta}}\mathbf{X}(0)$, and $\mathbf{F}(t) := \int_0^t e^{(t-s)\boldsymbol{\Delta}}\mathbf{Q}(s)\mathrm{d}s$.

**Interpretation of the heat equation.** The above heat equation can be interpreted as a continuous way of performing message passing in the graph $G$ over $t$ continuous steps by exploiting the neighborhood information included in $\boldsymbol{\Delta}$. Nonetheless, our continuous message passing approach introduces noise due to the presence of the nonhomogeneous term.

Each column of $\mathbf{N} = \mathbf{X}(0)$ contains some initial signal of the nodes, and the information of the Laplacian matrix and noise are jointly diffused in those signals as $t$ increases by following the heat equation in equation 2. One difference with standard diffusion processes is the use of the global topological information of the graph via $e^{t\boldsymbol{\Delta}}$ over time $t$. The noise is introduced via the term $\mathbf{F}(t)$. If $\mathbf{Q}$ is homogeneous, then $\forall t \geq 0, \mathbf{F}(t) = \mathbf{0}$ (i.e., there is no noise) and equation 2 reduces to $\forall t \geq 0, \mathbf{X}(t) = \mathbf{Z}(t)$.

## 3 Heat Diffusion Encoder

As in standard denoising autoencoders (Vincent et al., 2008), we define a noising process that maps $\mathbf{X}(0)$ to an informative representation close to some analytically tractable distribution (from which we could sample at inference time). To this end, we consider that equation 2 is the output of our *heat diffusion encoder* that is given $\mathbf{X}(0)$ and $\boldsymbol{\Delta}$ as input, and it constructs some noisy representation $\mathbf{X}(T)$ where $T > 0$ is an arbitrary time constant defined in Section 4. In Proposition 1, we propose to define the heat source term $\mathbf{Q}$ so that $\mathbf{X}(T)$ is *similar* to some constant matrix that we call $\mathbf{M}$ and that has all its elements equal to the same value in order to approximate maximum entropy (or lack of information) in the node representations when $t = T$. In Section 4, we train a decoder that reconstructs the nodes and edges when given some noisy $\mathbf{X}(T)$.

**Formulation of the heat source term.** Our goal is to formulate the heat source term $\mathbf{Q}$ so that $\mathbf{X}(T)$ tends to some non-informative matrix $\mathbf{M}$ as $T$ tends to $+\infty$. To this end, we first notice that the matrix $e^{t\boldsymbol{\Delta}}$ is column stochastic for all $t \geq 0$ (see explanation in Appendix F). In order to have $\mathbf{Z}(t) = e^{t\boldsymbol{\Delta}}\mathbf{N}$ also column stochastic for all $t \geq 0$, we add the constraint that $\mathbf{N} = \mathbf{X}(0)$ is column stochastic (see proof in Appendix C.3). Since each column of the resulting node representations is a probability distribution vector, we define each column of $\mathbf{M}$ as the uniform probability distribution vector, which corresponds to the maximum entropy probability distribution vector. In other words, we define the column stochastic uniform noise matrix as $\mathbf{M} := \frac{1}{n}\mathbf{1}\mathbf{1}^{\top} \in \{\frac{1}{n}\}^{n \times d}$. By definition, the matrix $\mathbf{M}$ is constant. Each column of $\mathbf{M}$ also corresponds to the expected value of a random variable following a flat Dirichlet distribution.

**Proposition 1.** *To satisfy* $\lim_{T \to +\infty} \mathbf{X}(T) = \mathbf{M}$ *where* $\mathbf{M} = \frac{1}{n}\mathbf{1}\mathbf{1}^{\top}$, *we formulate* $\mathbf{Q}$ *and* $\mathbf{F}$ *as follows:*

$$\mathbf{Q}(s) := \alpha e^{-\alpha s} e^{s\boldsymbol{\Delta}} \left(\mathbf{R} - e^{\beta\boldsymbol{\Delta}}\mathbf{X}(0)\right) \implies \mathbf{F}(t) = (1 - e^{-\alpha t})e^{t\boldsymbol{\Delta}} \left(\mathbf{R} - e^{\beta\boldsymbol{\Delta}}\mathbf{X}(0)\right) \tag{3}$$

*where* $\alpha > 0$ *is a noise diffusivity rate hyperparameter,* $\beta \geq 0$ *is a hyperparameter that can be tuned to control the Laplacian dynamics further, and* $\mathbf{R} := e^{-T\boldsymbol{\Delta}}\mathbf{M} \in \mathbb{R}^{n \times d}$ *is a constant matrix for some arbitrary time constant* $T > 0$ *defined in Section 4. See Appendix C for details and proofs.*

With the above proposition, $\mathbf{X}(t)$ can be written only as a function of $\mathbf{X}(0)$ and $\boldsymbol{\Delta}$ in equation 2:

$$\forall t \geq 0, \mathbf{X}(t) = e^{t\boldsymbol{\Delta}} \left(\mathbf{X}(0) + (e^{-\alpha t} - 1)e^{\beta\boldsymbol{\Delta}}\mathbf{X}(0) + (1 - e^{-\alpha t})\mathbf{R}\right) \tag{4}$$

If we set $\beta = 0$, then $e^{\beta\boldsymbol{\Delta}} = \mathbf{I}$ by definition of the matrix exponential, and we get a simpler formulation of equation 4:

$$\beta = 0 \implies \forall t, \mathbf{X}(t) = e^{-\alpha t}\mathbf{Z}(t) + (1 - e^{-\alpha t})e^{t\boldsymbol{\Delta}}\mathbf{R}.$$

In this case, we have the following simple formulation when $t = T$:

$$\beta = 0 \implies \mathbf{X}(T) = e^{-\alpha T}\mathbf{Z}(T) + (1 - e^{-\alpha T})\mathbf{M} \tag{5}$$

which is column stochastic, and we call $1 - e^{-\alpha T}$ the *noise ratio* at time $T$. Since we have $\lim_{T \to +\infty} e^{-\alpha T} = 0$, one can verify that Proposition 1 satisfies our initial goal because equation 5 implies $\lim_{T \to +\infty} \mathbf{X}(T) = \mathbf{M}$.

**Backward formulation.** In the following, we consider that $\beta = 0$. However, our approach can be generalized to any $\beta \geq 0$. If $\boldsymbol{\Delta}$ is given, we can write $\mathbf{X}(t)$ as a function of $\mathbf{X}(t + \tau)$ for all time step $\tau \geq 0$:

$$\beta = 0 \implies \forall t \geq 0, \mathbf{X}(t) = e^{\alpha\tau}e^{-\tau\boldsymbol{\Delta}}\mathbf{X}(t + \tau) + (1 - e^{\alpha\tau})e^{t\boldsymbol{\Delta}}\mathbf{R} \tag{6}$$

However, we assume that the denoising decoder that we train in Section 4 does not have access to $\mathbf{\Delta}$ at inference time. Therefore, the decoder has to reconstruct the set of nodes and/or edges when given only $\mathbf{X}(T)$.

**Efficient forward process.** As explained in Vignac et al. (2023), we can build an efficient generative model for the following reasons:

• The noisy representation $\mathbf{X}(t)$ has a closed-form expression that depends only on $\mathbf{X}(0)$ and $\mathbf{\Delta}$ (see equation 4). We can then directly calculate $\mathbf{X}(t)$ for all $t \geq 0$ without adding noise iteratively.

• The ground truth node representation $\mathbf{Z}(t)$ can be written in closed-form when given only $\mathbf{\Delta}$ and either $\mathbf{X}(0)$ or $\mathbf{X}(t)$. In practice, we generate a perturbed adjacency matrix $\tilde{\mathbf{A}}$, from which we generate its Laplacian $-\tilde{\mathbf{\Delta}}$ and noisy representation $\tilde{\mathbf{X}}(T) = e^{-\alpha T}e^{t\tilde{\mathbf{\Delta}}}\mathbf{X}(0) + (1 - e^{-\alpha T})\mathbf{M}$ that is given as input of a node decoder that has to reconstruct the ground truth node representation $\mathbf{Z}(t) = e^{t\mathbf{\Delta}}\mathbf{X}(0)$ for some $t \in [0, T]$. This data augmentation technique acts as a regularizer that reconstructs node representations similar to the training distribution when given a noisy input.

• The limit distribution $\lim_{T \to +\infty} \mathbf{X}(T) = \mathbf{M}$ does not depend on $\mathbf{X}(0)$. It is also worth noting that all the elements of $\mathbf{X}(T)$ are in the interval $\left[\left(1 - e^{-\alpha T}\right)/n, \left(1 + (n-1)e^{-\alpha T}\right)/n\right]$. In practice, we choose appropriate values of $T > 0$ and $\alpha > 0$ so that sufficient information of the graph is preserved in the noisy node representation, and so that denoised edges can be recovered from it. Specifically, we diffuse toward a distribution that can be well approximated by an analytic distribution (e.g., we can sample from a (flat) symmetric Dirichlet distribution (Kotz et al., 2004)) while preserving sufficient information about $\mathbf{X}(0)$ to perform denoising. Moreover, $\mathbf{X}(t)$ is column stochastic when $t = 0$ and $t \geq T$, but $\mathbf{X}(t)$ might contain some negative elements when $t \in (0, T)$ due to the formulation of $\mathbf{R}$. This is not a problem in practice since our goal is to reconstruct $\mathbf{Z}(t)$ which is column stochastic for all $t \geq 0$.

## 4 Denoising Decoder and Sampling

The previous section defines a noising *heat diffusion encoder* that does not require learning a neural network and is given an initial node representation matrix $\mathbf{N} = \mathbf{X}(0)$ and a Laplacian matrix to generate in closed form some noisy representation at some arbitrary time $T > 0$. We now propose a multi-task learning formulation to train decoders that reconstruct the original input. We assume that the decoders are not directly given the Laplacian matrix at inference time.

Our first task learns a neural network (called *node decoder*) that predicts the denoised node representation $\mathbf{Z}(t)$ where $t \in [0, T]$ (we recall that $\mathbf{Z}(0) = \mathbf{X}(0)$). Our second and last task jointly learns another neural network (called *edge decoder*) to predict edges. This is similar to the way link prediction tasks are solved, and we observe in practice that the learned representations hold information from the graph in the form of the Laplacian singular vectors. In Section 4.3, we propose a sampling strategy to generate digraphs at inference time.

### 4.1 Training of the denoising decoders and data augmentation

**Setup.** We consider the task where, during training, we are given $m$ digraphs $\{G_i = (V_i, E_i)\}_{i=1}^m$ drawn from some distribution $\mathcal{G}$ and of different sizes. Our goal is to generate graphs that follow the same distribution. Each graph $G_i$ is represented by its adjacency matrix $\mathbf{A}^i \in \{0, 1\}^{n_i \times n_i}$ where $n_i := |V_i|$ is the number of nodes. The Laplacian matrix of $G_i$ is $-\mathbf{\Delta}^i = \mathbf{I} - \mathbf{A}^i \left(\text{diag}(\mathbf{1}^\top \mathbf{A}^i)\right)^{-1}$.

**Node representation.** We explain here how we define the node representations of the training graphs. Let us note $n_{\max} := \max_i n_i$ the number of nodes of the largest graph in the training set. We define some matrix $\mathbf{O} \in \mathbb{R}^{n_{\max} \times d}$ where $d > 0$ is an arbitrary hyperparameter. For each graph $G_i$, we define its column stochastic initial node representation matrix $\mathbf{N}^i = \mathbf{X}^i(0) \in [0, 1]^{n_i \times d}$ as the uppersubmatrix of $\mathbf{O}$ whose columns are $\ell_1$-normalized with the softmax operator, which corresponds to applying a mask and renormalizing. To simplify the notation, we write $\mathbf{N}$ instead of $\mathbf{N}^i$ since we consider that all the graphs of same size share the same initial node representation. $\mathbf{N}$ is trained by training $\mathbf{O}$ (see details about their training in Appendix B).

---

**Algorithm 1** Generation of digraphs at inference time

---

**Input:** Node representations $\mathbf{O} \in \mathbb{R}^{n_{\max} \times d}$, hyperparameters $T, \alpha > 0$, $\mu, \rho \in [0, 1]$

1: Sample $n \leq n_{\max}$. Define $\mathbf{N} \in \mathbb{R}^{n \times d}$ as upper submatrix of $\mathbf{O}$ followed by softmax operation for $\ell_1$-normalized columns
2: Generate discrete adjacency matrix $\mathbf{A} \in \{0, 1\}^{n \times n}$ such that $\forall i \neq j$, $\mathbf{A}_{ij} \sim \text{Bernoulli}(\mu)$ and $\forall i$, $\mathbf{A}_{ii} = 1$
3: Apply data augmentation to obtain perturbed matrix $\tilde{\mathbf{A}}$ (e.g., $\tilde{\mathbf{A}} = \mathbf{A} \oplus \mathbf{C}$ s.t. $\forall i \neq j$, $\mathbf{C}_{ij} \sim \text{Bernoulli}(\rho)$)
4: Calculate the diagonal matrix $\mathbf{D} \in \mathbb{R}_+^{n \times n}$ such that $\mathbf{D}_{ii} = \sum_j \tilde{\mathbf{A}}_{ji}$. Define $\tilde{\boldsymbol{\Delta}} := \tilde{\mathbf{A}}\mathbf{D}^{-1} - \mathbf{I}$.

5: Define $\mathbf{B}$ as $e^{T\tilde{\boldsymbol{\Delta}}e}$ or optionally as the rank-$s$ approximation of $e^{T\tilde{\boldsymbol{\Delta}}}$ via truncated SVD.
6: Give the matrix $e^{-\alpha T}\mathbf{B}\mathbf{N} + (1 - e^{-\alpha T})\mathbf{M}$ as input of the edge decoder that returns an adjacency matrix

---

We arbitrarily define the values of $T > 0$ and $\alpha > 0$ so that the noise ratio defined as $(1 - e^{-\alpha T})$ is close to 1 (see Fig. 2). Following equation 5, we define $\mathbf{X}^i(T) := e^{-\alpha T}e^{T\boldsymbol{\Delta}^i}\mathbf{N} + (1 - e^{-\alpha T})\mathbf{M}$. In practice, we apply data augmentation on $\mathbf{X}^i(T)$ during training as explained below.

**Edge perturbation (Ding et al., 2022).** One can apply data augmentation via *edge perturbation* which can be interpreted as injecting noise by considering the perturbed adjacency matrix $\tilde{\mathbf{A}}^i = \mathbf{A}^i \oplus \mathbf{C}$ instead of $\mathbf{A}^i$, where $\oplus$ is the logical XOR operator, and the zero-diagonal corruption matrix $\mathbf{C} \in \{0, 1\}^{n_i \times n_i}$ has its non-diagonal elements equal to 1 with probability $\rho \in [0, 1]$, and 0 with probability $(1 - \rho)$. Following Veličković et al. (2019), we set $\rho \approx 1/n_i$ and we sample a new matrix $\mathbf{C}$ each time $\mathbf{C}$ is called. We call $-\tilde{\boldsymbol{\Delta}}^i := \mathbf{I} - \tilde{\mathbf{A}}^i(\text{diag}(\mathbf{1}^\top \tilde{\mathbf{A}}^i))^{-1}$ the Laplacian from $\tilde{\mathbf{A}}^i$. We obtain the formulation $\tilde{\mathbf{Z}}^i(T) := e^{T\tilde{\boldsymbol{\Delta}}^i}\mathbf{N}$, and $\tilde{\mathbf{X}}^i(T) := e^{-\alpha T}\tilde{\mathbf{Z}}^i(T) + (1 - e^{-\alpha T})\mathbf{M}$. If $\rho = 0$, we have $\tilde{\mathbf{X}}^i(T) = \mathbf{X}^i(T)$.

**Permutation of adjacency matrices.** $\mathbf{N}$ is the same for all graphs of same size. To promote permutation invariance of our model, we can replace $\mathbf{A}^i$ and $\boldsymbol{\Delta}^i$ by $\mathbf{P}^\top \mathbf{A}^i \mathbf{P}$ and $\mathbf{P}^\top \boldsymbol{\Delta}^i \mathbf{P}$ where $\mathbf{P} \in \{0, 1\}^{n_i \times n_i}$ is a (randomly sampled) permutation matrix. This is equivalent to replacing $e^{T\boldsymbol{\Delta}^i}\mathbf{N}$ by $\mathbf{P}^\top e^{T\boldsymbol{\Delta}^i}\mathbf{P}\mathbf{N}$. This can be seen as augmenting the training set with adjacency matrices of isomorphic digraphs. In Appendix G.3, we experimentally show that using this kind of data augmentation technique does not have a negative impact on the optimization of our loss function.

$\tilde{\mathbf{X}}^i(T)$ is given as input of a node decoder $\varphi$ and edge decoder $\psi$ during training as described below.

**Node decoding task.** Our node decoder $\varphi$ takes the noisy node representation $\tilde{\mathbf{X}}^i(T)$ as input, and its goal is to reconstruct some target node representation matrix $\mathbf{T}^i$ that does not contain noise. In practice, we formulate the training loss of our node decoder as $\mathscr{L}_{\text{node}}(i) := \|\varphi(\tilde{\mathbf{X}}^i(T)) - \mathbf{T}^i\|_F^2$ where $\|\cdot\|_F$ is the Frobenius norm, and we arbitrarily define $\mathbf{T}^i := \mathbf{Z}^i(1) = e^{\boldsymbol{\Delta}^i}\mathbf{N}$ where $-\boldsymbol{\Delta}^i$ is the ground truth Laplacian matrix. Since each row of $\tilde{\mathbf{X}}^i(T)$ represents a node of the graph, we ideally want our model to be equivariant to the order of the rows of $\tilde{\mathbf{X}}^i(T)$. For this reason, we formulate $\varphi$ as an attention-based permutation-invariant neural network called *Set Transformer* (Lee et al., 2019). $\varphi$ considers each row of $\tilde{\mathbf{X}}^i(T)$ as the element of a set of node representations, and it is robust to the order of the rows. Implementation details can be found in Appendix B.

**Edge decoding task.** We call our edge decoder $\psi$ and we denote the $p$-th row of $\psi(\tilde{\mathbf{X}}^i(T))$ by $\psi(\tilde{\mathbf{X}}^i(T))_p$. Our edge decoder predicts whether or not there exists a directed edge between pairs of nodes. We formulate the term: $\mathscr{L}_{\text{edge}}(i) := \sum_{p \neq q} H\left(\omega\left(\left[\psi(\tilde{\mathbf{X}}^i(T))_p, \psi(\tilde{\mathbf{X}}^i(T))_q\right]\right), \mathbf{A}_{pq}^i\right)$ where $[\cdot, \cdot]$ denotes concatenation, $\omega$ is a learned multilayer perceptron (MLP), and $H$ is the cross-entropy loss. Our edge decoder and node decoder share a common backbone (see architecture details in Appendix B). It is worth noting that if the goal is to generate undirected graphs, then the concatenation operation can be replaced by a symmetric operation such as the addition. The training loss that we minimize is:

$$\sum_{i=1}^m \mathscr{L}_{\text{edge}}(i) + \gamma \, \mathscr{L}_{\text{node}}(i) \tag{7}$$

where $\gamma \geq 0$ is a regularization parameter. Since both $\mathbf{T}^i$ and $\tilde{\mathbf{X}}^i(T)$ depend on $\mathbf{N}$, we optimize equation 7 by training jointly $\varphi$, $\psi$ and $\mathbf{N}$ via gradient descent in order to reconstruct the training graphs. See Appendix B for implementation details.

## 4.2 Other considerations

**Class-conditional generation.** To add class label information, we give as input of both decoders the concatenation of a matrix $\mathbf{Y}^i \in \{0,1\}^{n_i \times |C|}$ to $\tilde{\mathbf{X}}^i(T)$ where $|C|$ is the number of categories, and each row of $\mathbf{Y}^i$ is a one-hot vector whose nonzero index corresponds to the category of the graph $G_i$. This sampling strategy is known as conditional sampling (Zhu et al., 2022). The rest of the method is similar.

**Choice of the final step** $T$. The matrix $\tilde{\mathbf{X}}^i(T)$ is given as input of decoders to reconstruct $G_i$. We ideally want $\tilde{\mathbf{X}}^i(T) = e^{-\alpha T}\tilde{\mathbf{Z}}^i(T) + (1 - e^{-\alpha T})\mathbf{M}$ to be similar to the matrix $\mathbf{M}$. This similarity depends on both $T$ and $\alpha$, and $\tilde{\mathbf{X}}^i(T)$ tends to $\mathbf{M}$ as $T$ or $\alpha$ tend to $+\infty$. We provide a detailed discussion about the impact of $T$ and $\alpha$ in Appendix F. We found that setting $T = 1$ and choosing $\alpha$ large enough works well in practice (e.g., $\alpha = 2.3$ implies $1 - e^{-\alpha T} \approx 0.9$, which means that about 90% of the values of $\tilde{\mathbf{X}}^i(T)$ are noise). However, the optimal value of both $T$ and $\alpha$ can be determined via cross-validation depending on the task. Fig. 2 illustrates the ratio of noise for different values of $\alpha$ as a function of $T$. It is worth noting that we want $\tilde{\mathbf{X}}^i(T)$ to be similar to $\mathbf{M}$ so that sampling a similar matrix at inference time is easy. On the other hand, we also want $\tilde{\mathbf{X}}^i(T)$ to preserve enough information so that our neural networks can reconstruct $\mathbf{T}^i$ and $\mathbf{A}^i$ (i.e., the node and edge information of the graph) from it.

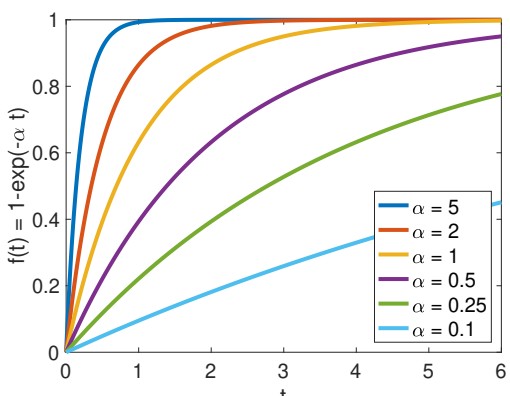

Figure 2: The noise ratio $1 - e^{-\alpha T}$ in $\mathbf{X}(T)$ as a function of $T$ for different values of $\alpha$.

**Learning N.** It is known in the heat kernel literature (Belkin & Niyogi, 2003; Chung & Yau, 1999) that the coarse structure of an undirected graph is included in the subset of eigenvectors of its Laplacian matrix that correspond to its smallest eigenvalues and can be used for dimensionality reduction. PageRank (Page, 1998) uses the same motivation and exploits the leading eigenvector of a nonsymmetric stochastic matrix. Spectre (Martinkus et al., 2022) exploits this observation by generating a symmetric Laplacian matrix spanned by a set of leading eigenvectors with bounded real eigenvalues. Since our eigenvalues and eigenvectors are usually complex and not unitary (when the adjacency matrix is not symmetric), we consider related linear algebra properties such as column spaces and singular vectors, which allow us to work with real values. We experimentally observe in Section 6.1 that the leading singular vectors of the learned matrix $\mathbf{Z}^i(t) = e^{t\mathbf{\Delta}^i}\mathbf{N}$ and of $e^{t\mathbf{\Delta}^i}$ tend to be strongly correlated, which suggests that our model learns $\mathbf{N}$ so that it maps to the leading left singular vectors of $e^{t\mathbf{\Delta}^i}$. This observation is mainly qualitative. We also observed that using a large number of columns $d$ to represent $\mathbf{N}$ helps in practice to recover edges.

**Approximations.** We show in Appendix E that our approach corresponds to a non-symmetric heat kernel method induced in a Reproducing Kernel Banach Space (RKBS) with a column stochastic and non-symmetric kernel matrix $\mathbf{K} = e^{-\alpha T}e^{T\mathbf{\Delta}^i} + \frac{1 - e^{-\alpha T}}{n_i}\mathbf{1}\mathbf{1}^\top$. $\mathbf{K}$ has the same eigenvectors as the Laplacian $-\mathbf{\Delta}^i$ when $\alpha = 0$ (see Appendix F), and it then contains the structure of the graph. To scale our method to large graphs, we also propose to replace $e^{t\mathbf{\Delta}^i}$ by its rank-$s$ approximation obtained with truncated Singular Value Decomposition (SVD) where $s \leq n_i$. We replace $e^{t\mathbf{\Delta}^i}$ by the product of two rectangular rank-$s$ matrices, which greatly reduces memory if $s \ll n_i$. Although the obtained matrix is not stochastic when $s < n_i$, we observe in Section 6.2 that the training graphs can be fully reconstructed with our model while saving memory usage. The SVD can be preprocessed offline before training. Gaussian kernels are specific heat kernels that use a squared Euclidean distance, and the relationship between heat kernels and Gaussian Processes (GPs) is known in the literature (Kondor & Lafferty, 2002). Using a truncated SVD corresponds to a low-rank approximation of the kernel matrix *w.r.t.* the Frobenius norm, which is similar to one of the approximation methods for GPs mentioned in Williams & Rasmussen (2006, Chapter 8.1).

### 4.3 Sampling strategy for generation of novel digraphs

Our sampling algorithm to generate novel digraphs is given in Algorithm 1 while our training algorithm is detailed in Algorithm 2. During sampling at inference time, we do not have access to input graphs from the dataset. We then construct graphs that, when given to our heat diffusion encoder, diffuse toward noisy graphs similar to those encountered after diffusing graphs during training. In this way, our denoising decoders can successfully produce denoised graphs similar to those of the dataset. This is conceptually similar to variational autoencoders (Kingma & Welling, 2014), where during sampling the encoding distribution is approximated by a simple prior distribution. How can we then analytically construct suitable input graphs during inference time? One solution is to generate a matrix with each column sampled from a flat Dirichlet distribution (Kotz et al., 2004) and give it as input of the decoders to generate a digraph. This works well when the training graphs (of a given category) are all similar to each other. However, it was observed in Vignac et al. (2023) that this kind of continuous sampling tends to destroy the graph's sparsity and creates very noisy graphs in practice. When the distribution of the graphs is multimodal, we found that sampling discrete adjacency matrices and applying standard data augmentation techniques for graphs both during training and sampling allows our model to sample graphs from the different modes. We study continuous sampling strategies in Appendix G.2.

We propose one discrete sampling algorithm in Algorithm 1. Let us note $\mu \in (0, 1]$ the ratio of pairs of distinct nodes that are adjacent in the training set. We first generate an adjacency matrix $\mathbf{A} \in \{0, 1\}^{n \times n}$ such that each of its non-diagonal elements is assigned the value 1 with probability $\mu$, and 0 otherwise. Following the motivation of denoising autoencoders, our decoders are trained to construct an (unperturbed) sample similar to training samples when given some noisy input. In Appendix G.2, we observe that our discrete sampling strategy is competitive with other continuous sampling strategies in terms of performance.

## 5 Related Work

Graph generative approaches can be divided into two categories which are *auto-regressive* models and *one-shot* models. Auto-regressive models (Liao et al., 2019; You et al., 2018) generate a succession of graphs $G_1, G_2, \ldots, G_T$ such that $\forall i, G_i \subset G_{i+1}$ and return the last generated graph $G_T$. At each iteration, the graph $G_i$ is given as input of a neural network that generates $G_{i+1}$ by adding new nodes and their edges. Most of these models are typically slower than one-shot approaches that generate all the nodes and edges of a graph in a single step. Three main one-shot approaches in the machine learning literature are Top-n (Vignac & Frossard, 2022), Spectre (Martinkus et al., 2022) and DiGress (Vignac et al., 2023). Other one-shot methods such as (Kwon et al., 2020; Mercado et al., 2021) are dedicated to molecular graphs and do not generalize to other tasks. Although Top-n is one-shot, it assumes symmetric similarity functions between nodes inappropriate for digraphs.

Spectre (Martinkus et al., 2022) considers the generation of undirected graphs via their normalized graph Laplacian matrix $\mathbf{L}_n := \mathbf{I} - \mathbf{D}^{-1/2}\mathbf{A}\mathbf{D}^{-1/2}$, which is symmetric positive semi-definite and admits an eigendecomposition of the form $\mathbf{U}^{-1}\mathbf{\Lambda}\mathbf{U}$ where $\mathbf{U}^{-1} = \mathbf{U}^\top$ and both the diagonal matrix $\mathbf{\Lambda}$ and $\mathbf{U}$ are real-valued. They exploit the intuition that coarse structure of the graph lies on a Stiefel manifold that contains the eigenvectors of the $k$ smallest eigenvalues of the Laplacian. Spectre Martinkus et al. (2022) then trains a neural network that generates an adjacency matrix by sampling the $k$ smallest eigenvalues and their corresponding eigenvectors by exploiting their Stiefel manifold structure. The authors mention that their work can be extended to the random-walk Laplacian matrix $\mathbf{L}_r := \mathbf{I} - \mathbf{A}\mathbf{D}^{-1}$ since its right eigenvectors are the same as $\mathbf{D}^{-1/2}\mathbf{U}$ (up to column-wise $\ell_2$ normalization), and its left eigenvectors can be formulated in a similar way. However, when the graph is directed and $\mathbf{A}$ is not symmetric, $\mathbf{U}$ is complex and not unitary. The information of the Laplacian matrix then does not lie on a complex Stiefel manifold, and Spectre can then not easily be extended to digraphs.

DiGress (Vignac et al., 2023) is a denoising diffusion model for graphs. Instead of using the discrete Laplacian operator as we propose, they represent their nodes as a function of time $t \in \mathbb{N}$ as follows: $\mathbf{X}(t) = (\alpha^t \mathbf{I} + \frac{(1-\alpha^t)}{d}\mathbf{1}\mathbf{1}^\top)\mathbf{X}(t-1)$ where $\mathbf{X}(t)$ is row-stochastic and $\alpha \in (0, 1)$. DiGress relies on spectral

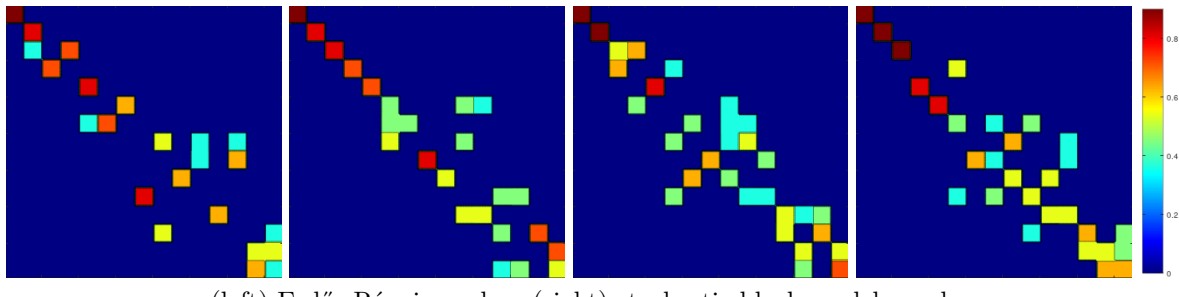

(left) Erdős-Rényi graphs    (right) stochastic block model graphs

Figure 3:   Correlations between the left singular vectors of $e^{t\boldsymbol{\Delta}^i}$ and $e^{t\boldsymbol{\Delta}^i}\mathbf{N}$ (more results in Fig. 4).

features from Beaini et al. (2021) designed for undirected graphs by using a symmetric similarity function between nodes. It is then not appropriate for digraphs.

Diffusion processes have been used in score-based generative models (Ho et al., 2020; Sohl-Dickstein et al., 2015; Song et al., 2021) to generate undirected graphs (Jo et al., 2022; Niu et al., 2020) or adjacency matrices in general (Yan et al., 2024). Nonetheless, both Jo et al. (2022) and Niu et al. (2020) exploit Graph Neural Network architectures that require a symmetric adjacency matrix so these methods are not easily adaptable to digraphs. Moreover, both approaches perturb each data dimension independently (i.e., without accounting for the global structure of the graph via its Laplacian). On the other hand, SwinGNN (Yan et al., 2024) considers diffusion models to reconstruct arbitrary adjacency matrices that do not have to be symmetric. Generally, in such score-based generative diffusion models, the input converges towards an entirely uninformative pure noise distribution, and generation requires a slow and iterative denoising process. In contrast, our approach encodes the global structure of the graph into the node representations via the Laplacian dynamics and effectively encodes the graph topology in a noisy, yet informative distribution when $t = T$, from which the denoised nodes and edges can be efficiently predicted in one shot. In other words, in contrast to score-based diffusion models, where the role of the diffusion process is purely to gradually perturb the data and destroy information, in our model the primary role of the (Laplacian) diffusion process is to additionally encode the graph structure into the node representations. Heat diffusion for the purpose of information propagation in that fashion has, for instance, been used for learning on curved 3D surfaces (Sharp et al., 2022). Our method is a different approach compared to the widely used score-based diffusion models and specifically designed to avoid their slow, iterative synthesis process.

Instead, our model can be seen as a denoising autoencoder (Vincent et al., 2008), since it corrupts the input data with optional edge perturbation and with a nonhomogoneous *heat diffusion* process. During sampling, we also approximate the encoding distribution at $t = T$ by using an analytically tractable distribution, which is similar to variational autoencoders (Kingma & Welling, 2014), where a simple prior distribution in latent space models the encoding distribution. Hence, our approach can be seen as related to GraphVAE (Simonovsky & Komodakis, 2018), which, however, uses learned encoder neural networks instead of a heat diffusion process to encode small graphs in a latent space. Moreover, GraphVAE only tackles undirected graph synthesis. We emphasize that connection by proposing a sampling method that directly samples from a flat Dirichlet distribution in Appendix G.2.

It is worth noting that it was shown in Vincent (2011) that score matching techniques (Hyvärinen & Dayan, 2005) can be seen as training a specific type of denoising autoencoder. Our approach could be adapted to a score-based generative model with iterative synthesis. This could be done by replacing our deterministic nonhomogeneous heat source term by a Wiener process using a Dirichlet diffusion score as done for example in Avdeyev et al. (2023). However, the method would be computationally expensive for a large number of nodes since (1) each node would be associated with a different beta distribution, (2) there exists no known closed-form solution for this kind of linear stochastic differential equation, (3) and each intermediate time step would require calculating some matrix exponential involving $\boldsymbol{\Delta}$ in our case. Instead, we propose a framework similar to Martinkus et al. (2022) to efficiently work with singular vectors of the Laplacian matrix and generate (directed) graphs in a one-shot manner.

## 6  Experiments

We evaluate the generative power of our method that we call Digraph Generation with Diffusion Kernels (DGDK), focusing on digraphs only, for which our method was designed. We use Adam (Kingma & Ba, 2014) as optimizer. DGDK works better as the number of columns $d$ of $\mathbf{N}$ is larger. Experimental details can be found in Appendix B. Due to lack of space, we add other experiments in Appendix G.

### 6.1  Column space of the learned representations

We adapt the intuition of Spectre (Martinkus et al., 2022) to digraphs. To this end, we experimentally observe in this subsection that our learned representations are strongly correlated with the leading singular vectors of $e^{t\boldsymbol{\Delta}^i}$. In our first qualitative experiment, we apply conditional sampling (Zhu et al., 2022) and consider two categories of digraphs: Erdős-Rényi (Erdős et al., 1960) with $p = 0.6$, and stochastic block model (Holland et al., 1983) with 3 main blocks that are connected to each other by following the transition probability matrix:

$$\boldsymbol{\Pi} = \begin{pmatrix} 0.9 & 0.35 & 0.2 \\ 0.2 & 0.75 & 0.2 \\ 0.2 & 0.25 & 0.8 \end{pmatrix} \tag{8}$$

The above transition matrix $\boldsymbol{\Pi}$ between blocks cannot correspond to undirected graphs since undirected graphs would require that $\boldsymbol{\Pi}$ is symmetric. Each category contains $m = 100$ training graphs with $n = 15$ nodes each, and we set the number of columns of $\mathbf{N}$ to $d = 150$. The size of the first two blocks is $m = \lfloor \frac{n}{4} \rfloor$ each and the size of the last block is $n - 2m$. In this setup, our training graphs are directed and their adjacency matrices are not symmetric. As explained in Section 5, standard baselines such as Spectre cannot be exploited.

If we define $\mathbf{T}^i = e^{\boldsymbol{\Delta}^i} \mathbf{N} \in [0, 1]^{n \times d}$ (i.e., we consider $t = 1$), the goal of the node decoder in Section 4.1 is to reconstruct $\mathbf{T}^i$ whose column space is by definition included in the column space of $e^{\boldsymbol{\Delta}^i}$ (i.e., spanned by its columns). $\mathbf{T}^i$ is in general not a square matrix (i.e., $n \neq d$) and thus does not possess eigenvectors. We then consider its SVD $e^{t\boldsymbol{\Delta}^i} \mathbf{N} = \mathbf{U}_1 \boldsymbol{\Lambda}_1 \mathbf{V}_1^\top$, and we write the SVD of $e^{t\boldsymbol{\Delta}^i} = \mathbf{U}_2 \boldsymbol{\Lambda}_2 \mathbf{V}_2^\top$. The singular values of both $\boldsymbol{\Lambda}_1$ and $\boldsymbol{\Lambda}_2$ are ordered in descending order. Figure 3 illustrates the absolute values of the following matrix product $\mathbf{U}_2^\top \mathbf{U}_1$. Since both $\mathbf{U}_1$ and $\mathbf{U}_2$ have their columns $\ell_2$-normalized, each element of $\mathbf{U}_2^\top \mathbf{U}_1$ is the cosine between two singular vectors. A cosine of 0 indicates orthogonality, hence independence, whereas higher absolute values indicate (cosine) correlations. The top left part of each plot corresponds to the leading singular vectors whereas the bottom right corresponds to the singular vectors with lower singular values. As one can see, there is a strong cosine correlation between the leading singular vectors of $e^{t\boldsymbol{\Delta}^i}$ and of $e^{t\boldsymbol{\Delta}^i} \mathbf{N}$. This suggests that $\mathbf{N}$ is learned to preserve the most informative singular vectors of $e^{t\boldsymbol{\Delta}^i}$. We exploit this observation in Section 6.2 by working with a low-rank approximation of $e^{t\boldsymbol{\Delta}^i}$.

### 6.2  Using approximations of the target matrix

We evaluate the generative power of DGDK in the class-conditional generation task. We use a rank-$s$ approximation of $e^{T\boldsymbol{\Delta}^i}$ via a truncated SVD, and we formulate our target node matrix $\mathbf{T}^i = \mathbf{Z}^i(T)$ where $T = 1$ and $\alpha = 2.3$. We consider the case where the number of nodes is $\forall i, n_i = 21$, and $s = 15$. The two categories follow the same properties as in Section 6.1, and contain 3,000 non-isomorphic training graphs per category. During training, each mini-batch contains 10 graphs per category. Only the number of nodes per graph is different. DGDK manages to reconstruct all the edges of the training graphs when given noisy representations $\mathbf{X}^i(T)$. This shows the effectiveness of our low-rank approximation approach.

We sample 10,000 test graphs per category by using Algorithm 1 with class-conditional generation (i.e., we provide the desired category as input). None of the generated test graphs are isomorphic to one another nor to the training graphs. This means that our model obtains a uniqueness and novelty scores of 100%. We also report in Table 1 standard evaluation metrics based on the squared Maximum Mean Discrepancy (MMD) (O'Bray et al., 2022) between the training set and the test set. These evaluation metrics measure the *distance* between the training and test distributions *w.r.t.* some graph properties. We adapt the descriptor functions in O'Bray et al. (2022) to directed graphs.

Table 1: Squared MMD distances.

| Dataset | Erdős-Rényi ($p = 0.6$) | | | Stochastic block model (3 blocks) | | |
|---|---|---|---|---|---|---|
| MMD metric | Degree | Clustering | Spectrum | Degree | Clustering | Spectrum |
| DGDK (ours) | $1.1 \times 10^{-4}$ | $\mathbf{1.0 \times 10^{-3}}$ | $\mathbf{1.3 \times 10^{-5}}$ | $1.2 \times 10^{-4}$ | $\mathbf{2.0 \times 10^{-4}}$ | $\mathbf{6.2 \times 10^{-6}}$ |
| SwinGNN (Yan et al., 2024) | $1.3 \times 10^{-4}$ | $2.4 \times 10^{-2}$ | $4.2 \times 10^{-3}$ | $1.2 \times 10^{-4}$ | $3.7 \times 10^{-2}$ | $5.1 \times 10^{-3}$ |
| GRAN (Liao et al., 2019) | $1.5 \times 10^{-4}$ | $5.6 \times 10^{-2}$ | $1.9 \times 10^{-4}$ | $1.3 \times 10^{-4}$ | $8.3 \times 10^{-2}$ | $3.6 \times 10^{-4}$ |

**Degree distribution histogram.** Given a graph $G = (V, E)$, we create a $n$-dimensional histogram by evaluating the in-degree $\deg(v)$ for $v \in V$. The $i$-th position of the resulting histogram is the number of nodes with in-degree $i$. We $\ell_1$-normalize the histogram so that it sums to 1.

**Clustering coefficient.** The local clustering coefficient for a directed graph of a node $v_i$ is formulated: $C_i := \frac{|\{(v_j, v_k) \in E : v_j \in N_i, v_k \in N_i\}|}{|N_i|(|N_i| - 1)} \in [0, 1]$ where $N_i = \{v_j : (v_i, v_j) \in E \text{ or } (v_j, v_i) \in E\}$. It measures to what extent $v_i$ forms a clique. The different values $C_1, \ldots, C_n$ are binned into a $b$-dimensional histogram. We set $b = 100$.

**Laplacian spectrum.** In the directed case, the eigenvalues of $\mathbf{S} = \mathbf{L} + \mathbf{I}$ are complex but their absolute value is upper bounded by 1 since $\mathbf{S}$ is column stochastic. We bin their absolute values into a 100-dimensional histogram.

We report in Table 1 the scores of DGDK, GRAN (Liao et al., 2019) (the state-of-the-art auto-regressive baseline that can be extended to digraphs), and SwinGNN (Yan et al., 2024) (a diffusion approach that can be applied to arbitrary adjacency matrices). It is worth noting that we train a different GRAN and SwinGNN model for each category instead of using a class-conditional generation approach. Our MMD scores are close to 0, which means that the training and test graphs follow similar distributions. Moreover, DGDK outperforms both baselines in the clustering and spectrum evaluation metrics. This suggests that learning the global structure of the graph via its Laplacian in a single shot is beneficial for generation (for example compared to GRAN that sequentially considers multiple local subproblems).

### 6.3 Additional experiments

We present additional experiments in Appendix G. In Appendix G.2, we consider an experiment similar to the one in Section 6.2 with digraphs containing different numbers of nodes (from 180 to 200). We compare different sampling strategies and show that strategies that exploit the learned matrix $\mathbf{N}$ tend to perform better. In Appendix G.4, we consider a case where the training distribution contains multiple modes, and DGDK generates samples similar to the different modes.

## 7 Discussion

We discuss here different aspects of our approach that explained in detail in the appendix.

### 7.1 Scalability.

One main advantage of our approach is that we use efficient closed-form solutions for the encoder that can be preprocessed offline to obtain efficient training times. As explained in Appendix B, in practice on an NVIDIA GeForce RTX 3090 with 24 GB or VRAM, we manage to train 10,000 iterations of the dataset introduced in Section 6.2 in about 1 hour when the batch size is 20 (10 training graphs per category). We can also train in 1 hour 10,000 iterations of the dataset introduced in Section G.2, with larger graphs containing up to 200 nodes and a batch size of 2 (1 training graph per category). The main limitation to scale to larger graphs is memory. Indeed, our low-rank approximation technique presented in Section 4.2 allows us to save memory that grows linearly in the size of the training set. Nonetheless, larger graphs require larger VRAM.

## 7.2 Connection with Heat Kernels and Reproducing Kernel Banach Spaces (RKBS)

The understanding of heat kernels and RKBS is not essential to run our method. Nonetheless, it provides a connection with existing work that considers graphs and kernel methods (Kondor & Lafferty, 2002; Belkin & Niyogi, 2003). In particular, this connection is used in Belkin & Niyogi (2003) to create low-dimensional representations of the nodes, and in Kondor & Lafferty (2002) to perform classification by using the kernel matrix as a similarity measure between nodes. In both cases, their graphs are undirected and their kernel matrix is symmetric, which allows them to exploit Reproducing Kernel Hilbert Spaces (RKHS).

In our case, we propose to add a nonhomogeneous term to introduce noise, and we show that the resulting non-symmetric matrix $\mathbf{K}$ still corresponds to a non-symmetric kernel matrix induced by a RKBS. This motivates our low-rank approximation, inspired by kernel methods, to reduce used memory that grows linearly in the number of nodes instead of quadratically.

It is worth noting that since the kernel matrix $\mathbf{K}$ is non-symmetric, the standard framework of RKHS that relies on (symmetric) inner products cannot be used. Our RKBS interpretation introduces a elegant alternative that is useful both theoretically and in practice.

## 8 Conclusion and Limitations

We have proposed a one-shot generative model that samples digraphs and is similar in essence to denoising autoencoders. Our encoder exploits closed-form expressions to add noise to a digraph, and our decoder is trained to recover the global structure of the graph via its Laplacian dynamics. We show how our framework generalizes heat kernels and is able to simulate the training distribution. We also propose a low-rank approximation of the heat kernel matrix to possibly scale to large graphs.

### Limitations

Although our approach is scalable and produces digraphs of different sizes, one limitation is that our decoder is deterministic and its output is determined by its input. Nonetheless, different sampling strategies that we study thoroughly in Appendix G.2 can be used to generate a large set of different types of input. This allows our model to generate a diverse set of outputs.

Another limitation of our work is the lack of experiments on real-world datasets. This limitation is due to the fact that we did not find an appropriate real-world dataset to try our approach. For instance, even the genome dataset introduced in Marbach et al. (2012) contains three different graphs, each for a different type of gene. Our goal is to work with a large number of graphs that follow the same type of distribution. On the other hand, in order to work with the dataset in Marbach et al. (2012), we would need to generate multiple subgraphs from the three large graphs, which is different from the task we are interested in.

### Broader Impact Statement

Our work is not focused on applications. There are many potential societal consequences of our work, none which we feel must be specifically highlighted in the paper.

### Acknowledgments

We thank Nicholas Sharp for helpful discussions about this project and the anonymous reviewers for their feedback.

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

---

**Algorithm 2** Training algorithm (for each mini-batch)

---

**Input:** Node representations $\mathbf{O} \in \mathbb{R}^{n_{\max} \times d}$, hyperparameters $T > 0$, $\alpha > 0$, $t \geq 0$, Bernoulli factor $\rho \in [0, 1]$

  Initialize mini-batch loss as $\mathscr{L}_{\text{mini-batch}} = 0$
  **for** Graph $G_i$ with Adjacency matrix $\mathbf{A}^i$ and Laplacian matrix $\mathbf{\Delta}^i$ in the mini-batch **do**
    **if** Promote permutation invariance is *true* **then**
      Generate random permutation matrix $\mathbf{P} \in \{0, 1\}^{n_i \times n_i}$.
      $\mathbf{A}^i \leftarrow \mathbf{P}^\top \mathbf{A}^i \mathbf{P}$.
      $\mathbf{\Delta}^i \leftarrow \mathbf{P}^\top \mathbf{\Delta}^i \mathbf{P}$
    **end if**
    **if** Data augmentation matrix is *true* **then**
      Generate perturbation matrix $\mathbf{C} \in \{0, 1\}^{n_i \times n_i}$ s.t. $\forall i, \mathbf{C}_{ii} = 0$ and $\forall i \neq j, \mathbf{C}_{ij} \sim \text{Bernoulli}(\rho)$
      $\tilde{\mathbf{A}}^i \leftarrow \mathbf{A}^i \oplus \mathbf{C}$.
      $\tilde{\mathbf{\Delta}}^i \leftarrow \tilde{\mathbf{A}}^i \left(\text{diag}(\mathbf{1}^\top \tilde{\mathbf{A}}^i)\right)^{-1} - \mathbf{I}$
    **else**
      $\tilde{\mathbf{\Delta}}^i \leftarrow \mathbf{\Delta}^i$.
    **end if**
    Define $\mathbf{N} \in \mathbb{R}^{n \times d}$ as upper submatrix of $\mathbf{O}$ followed by softmax operation for $\ell_1$-normalized columns
    Define $\mathbf{B}$ as $e^{T \tilde{\mathbf{\Delta}}^i}$ or optionally as the rank-$s$ approximation of $e^{T \tilde{\mathbf{\Delta}}^i}$ via truncated SVD.
    $\mathbf{X}^i(T) \leftarrow e^{-\alpha T} \mathbf{B} \mathbf{N} + (1 - e^{-\alpha T}) \mathbf{M}$
    Define $\mathbf{E}$ as $e^{t \mathbf{\Delta}^i}$ or optionally as the rank-$s$ approximation of $e^{t \mathbf{\Delta}^i}$ via truncated SVD.
    $\mathbf{T}^i \leftarrow e^{-\alpha t} \mathbf{E} \mathbf{N}$
    $\mathscr{L}_{\text{mini-batch}} \leftarrow \mathscr{L}_{\text{mini-batch}} + \mathscr{L}_{\text{edge}}(i) + \gamma \, \mathscr{L}_{\text{node}}(i)$ where $\gamma \geq 0$ is a regularization parameter.
  **end for**
  Optimize $\mathscr{L}_{\text{mini-batch}}$ with Adam Kingma & Ba (2014)

---

# A  Summary

Our appendix is structured as follows:

• Section B provides experimental details including the architecture of the neural networks.

• Section C provides the details of our equations for our heat diffusion encoder.

• Section D provides the necessary details to solve the consensus problem.

• Section E explains the connection of our method with heat kernels.

• Section F studies the impact of $T$ and $\alpha$ on the eigenvalues of the different matrices that are involved in our model.

• Section G presents additional experimental results.

# B  Experimental details

**Setup.** We ran all our experiments on a single desktop with a NVIDIA GeForce RTX 3090 GPU (with 24 GB of VRAM) and 64 GB of RAM. We coded our project in Pytorch. We use double precision format to define our tensors. This is important in the current version of Pytorch to obtain an accurate version of the matrix exponential.

Our training algorithm is illustrated in Algorithm 2 and follows the different data augmentation techniques mentioned in Section 4.

**Node representation matrix N.** We define $n_{\max} \in \mathbb{N}$ as the maximum number of nodes in a graph of the training set, and $d$ as the number of columns of the initial node representation $\mathbf{O} \in \mathbb{R}^{n_{\max} \times d}$. In practice, we set $d = 150$ and we initialize each element of $\mathbf{O}$ by sampling from the normal distribution parameterized by a mean of 1 and standard deviation of 1. Other hyperparameter values could be used. For a given graph $G_i$ of $n_i$ nodes, we define $\mathbf{N} \in \mathbb{R}^{n_i \times d}$ as the upper submatrix of $\mathbf{O}$, and it is $\ell_1$-normalized by using a column-wise softmax operator. Equation 7 is minimized via standard gradient descent by training jointly $\psi$, $\varphi$ and $\mathbf{N}$

(and hence $\mathbf{O}$). In practice, we find that the larger $d$, the better the performance. However, our approach is limited by the amount of available memory.

In Section 6.2, we train the model for 60,000 iterations. We recall that each mini-batch contains 10 training graphs per category, hence we have 20 graphs per mini-batch. We make this choice because this is the maximum amount we manage to load in our GPU VRAM. The training algorithm takes about one hour for 10,000 iterations, so about 6 hours in total. We use a regularization parameter of $\gamma = 100$, and a step size/learning rate of 0.0001. Following O'Bray et al. (2022), we calculate our MMD evaluation metric by using a radial basis function (RBF) kernel with $\sigma = 10$. We report performance for other values of $\sigma$ in Appendix G.2.

**Backbone.** The common backbone of the node and edge decoder is:

- a linear layer $\mathbb{R}^d \to \mathbb{R}^{d'}$ where $d' = 1000$.

- a set transformer (Lee et al., 2019) with four attention blocks, each of dimensionality 1000, with one head, 20 inducing points and 20 seed vectors. Each row-wise feedforward layer of an attention block contains 3 linear layers $d' \times d'$ with ReLU activation function. We choose one head so that the global structure of the Laplacian matrix is not separately processed by the different heads.

- a multilayer perceptron (MLP) with one linear layer $\mathbb{R}^{d'} \to \mathbb{R}^{d''}$ (with $d'' = 600$), followed by four linear layers $\mathbb{R}^{d''} \to \mathbb{R}^{d''}$, and one linear layer $\mathbb{R}^{d''} \to \mathbb{R}^{d'''}$ where $d''' = 200$.

**The head of the node decoder** is an MLP with an initial layer $\mathbb{R}^{d'''} \to \mathbb{R}^{d''''}$ followed by one linear layer $\mathbb{R}^{d''''} \to \mathbb{R}^{d''''}$, and one linear layer $\mathbb{R}^{d''''} \to \mathbb{R}^d$. If the input is of size $\mathbb{R}^{n_i \times d}$, this returns a matrix of same size. We also use column-wise normalization with softmax on the output.

**Edge decoder.** The output of the backbone described above are node representations of size $d''' = 200$ for each node. We concatenate them (as described in Section 4.1) to obtain representations of pairs of nodes of size $2d''' = 400$. They are given as input of an MLP with one linear layer $\mathbb{R}^{2d'''} \to \mathbb{R}^{d''''''}$ (with $d'''''' = 600$), followed by 5 linear layers $\mathbb{R}^{d''''''} \to \mathbb{R}^{d''''''}$, followed by a linear layer $\mathbb{R}^{d''''''} \to \mathbb{R}^2$ that is used for cross-entropy loss (one element is used for the absence of edge, and the other element is used for the existence of edge). Alternatively, one could output a real value for each pair of nodes and use a binary cross entropy loss.

We use ReLU as an activation function between all the linear layers.

In all our experiments, we give the same weighting for the positive and negative edges (i.e., presence or absence of edge).

## C  Details of Equations for the Heat Diffusion Encoder

### C.1  Nonhomogeneous heat source term

We now give the details of the equations in Section 3. We recall the formulation of $\mathbf{Q}$ in Proposition 1:

$$\mathbf{Q}(s) = \alpha e^{-\alpha s} e^{s\boldsymbol{\Delta}} (\mathbf{R} - e^{\beta \boldsymbol{\Delta}} \mathbf{X}(0)) \tag{9}$$

which implies the following formulation of $\mathbf{F}$:

$$\mathbf{F}(t) := \int_0^t e^{(t-s)\boldsymbol{\Delta}} \mathbf{Q}(s) \mathrm{d}s = e^{t\boldsymbol{\Delta}} \int_0^t e^{-s\boldsymbol{\Delta}} \mathbf{Q}(s) \mathrm{d}s \tag{10}$$

$$= \int_0^t \alpha e^{(t-s)\boldsymbol{\Delta}} e^{s\boldsymbol{\Delta}} e^{-\alpha s} (\mathbf{R} - e^{\beta \boldsymbol{\Delta}} \mathbf{X}(0)) \mathrm{d}s = e^{t\boldsymbol{\Delta}} \int_0^t \alpha e^{-\alpha s} (\mathbf{R} - e^{\beta \boldsymbol{\Delta}} \mathbf{X}(0)) \mathrm{d}s \tag{11}$$

$$= e^{t\boldsymbol{\Delta}} \left( -e^{-\alpha s} (\mathbf{R} - e^{\beta \boldsymbol{\Delta}} \mathbf{X}(0)) \right) \Big|_{s=0}^{s=t} = (1 - e^{-\alpha t}) e^{t\boldsymbol{\Delta}} (\mathbf{R} - e^{\beta \boldsymbol{\Delta}} \mathbf{X}(0)) \tag{12}$$

## C.2 Node representation over time

We assume in this subsection that $t \geq 0$. equation 4 is written:

$$\mathbf{X}(t) = e^{t\boldsymbol{\Delta}}\left(\mathbf{X}(0) + (e^{-\alpha t} - 1)e^{\beta\boldsymbol{\Delta}}\mathbf{X}(0) + (1 - e^{-\alpha t})\mathbf{R}\right) \tag{13}$$

For any nonnegative time step $\tau \geq 0$, we can write $\mathbf{X}(t + \tau)$ as a function of $\mathbf{X}(t)$ and vice versa.

$$\mathbf{X}(t + \tau) = e^{(t+\tau)\boldsymbol{\Delta}}\left(\mathbf{X}(0) + (e^{-\alpha(t+\tau)} - 1)e^{\beta\boldsymbol{\Delta}}\mathbf{X}(0) + (1 - e^{-\alpha(t+\tau)})\mathbf{R}\right)$$

$$= e^{\tau\boldsymbol{\Delta}}\left(e^{t\boldsymbol{\Delta}}\left(\mathbf{X}(0) + (e^{-\alpha t} - 1)e^{\beta\boldsymbol{\Delta}}\mathbf{X}(0) + (1 - e^{-\alpha t})\mathbf{R} + (e^{-\alpha(t+\tau)} - e^{-\alpha t})(e^{\beta\boldsymbol{\Delta}}\mathbf{X}(0) - \mathbf{R})\right)\right)$$

$$= e^{\tau\boldsymbol{\Delta}}\left(\mathbf{X}(t) + \left(e^{-\alpha t} - e^{-\alpha(t+\tau)}\right)e^{t\boldsymbol{\Delta}}\left(\mathbf{R} - e^{\beta\boldsymbol{\Delta}}\mathbf{X}(0)\right)\right)$$

From the equation above, we find:

$$\mathbf{X}(t) = e^{-\tau\boldsymbol{\Delta}}\mathbf{X}(t + \tau) - \left(e^{-\alpha t} - e^{-\alpha(t+\tau)}\right)e^{t\boldsymbol{\Delta}}\left(\mathbf{R} - e^{\beta\boldsymbol{\Delta}}\mathbf{X}(0)\right) \tag{14}$$

When $\beta = 0$, we have:

$$\mathbf{X}(T) = e^{-\alpha T}e^{T\boldsymbol{\Delta}}\mathbf{X}(0) + (1 - e^{-\alpha T})e^{T\boldsymbol{\Delta}}\mathbf{R} = e^{-\alpha T}\mathbf{Z}(T) + (1 - e^{-\alpha T})\mathbf{M} \tag{15}$$

Let us assume that $\beta = 0$, equation 13 can be written as:

$$\mathbf{X}(t) = e^{t\boldsymbol{\Delta}}\left(e^{-\alpha t}\mathbf{X}(0) + (1 - e^{-\alpha t})\mathbf{R}\right) \tag{16}$$

which implies

$$\mathbf{X}(0) = e^{\alpha t}e^{-t\boldsymbol{\Delta}}\mathbf{X}(t) + (1 - e^{\alpha t})\mathbf{R} \tag{17}$$

Equation 17 implies the following formulation of $\mathbf{X}(0)$ as a function of $\mathbf{X}(t + \tau)$:

$$\mathbf{X}(0) = e^{\alpha(t+\tau)}e^{-(t+\tau)\boldsymbol{\Delta}}\mathbf{X}(t + \tau) + (1 - e^{\alpha(t+\tau)})\mathbf{R} \tag{18}$$

From equation 14, and by setting $\beta = 0$, we obtain:

$$\mathbf{X}(t) = e^{-\tau\boldsymbol{\Delta}}\mathbf{X}(t + \tau) - \left(e^{-\alpha t} - e^{-\alpha(t+\tau)}\right)e^{t\boldsymbol{\Delta}}\left(\mathbf{R} - \mathbf{X}(0)\right) \tag{19}$$

$$= e^{-\tau\boldsymbol{\Delta}}\mathbf{X}(t + \tau) + \left(e^{-\alpha(t+\tau)} - e^{-\alpha t}\right)e^{t\boldsymbol{\Delta}}\mathbf{R} + \left(e^{-\alpha t} - e^{-\alpha(t+\tau)}\right)e^{t\boldsymbol{\Delta}}\mathbf{X}(0) \tag{20}$$

By using equation 18, the last term of equation 20 can be rewritten:

$$\left(e^{-\alpha t} - e^{-\alpha(t+\tau)}\right)e^{t\boldsymbol{\Delta}}\mathbf{X}(0) = \left(e^{-\alpha t} - e^{-\alpha(t+\tau)}\right)e^{t\boldsymbol{\Delta}}\left(e^{\alpha(t+\tau)}e^{-(t+\tau)\boldsymbol{\Delta}}\mathbf{X}(t + \tau) + (1 - e^{\alpha(t+\tau)})\mathbf{R}\right) \tag{21}$$

$$= (e^{\alpha\tau} - 1)e^{-\tau\boldsymbol{\Delta}}\mathbf{X}(t + \tau) + \left(\left(e^{-\alpha t} - e^{-\alpha(t+\tau)}\right)\left(1 - e^{\alpha(t+\tau)}\right)e^{t\boldsymbol{\Delta}}\mathbf{R}\right) \tag{22}$$

Equation 20 is then rewritten:

$$\mathbf{X}(t) = e^{\alpha\tau}e^{-\tau\boldsymbol{\Delta}}\mathbf{X}(t + \tau) + \left(e^{-\alpha(t+\tau)} - e^{-\alpha t} + \left(e^{-\alpha t} - e^{-\alpha(t+\tau)}\right)\left(1 - e^{\alpha(t+\tau)}\right)\right)e^{t\boldsymbol{\Delta}}\mathbf{R} \tag{23}$$

$$= e^{\alpha\tau}e^{-\tau\boldsymbol{\Delta}}\mathbf{X}(t + \tau) + \left(e^{-\alpha(t+\tau)} - e^{-\alpha t} + e^{-\alpha t} - e^{\alpha\tau} - e^{-\alpha(t+\tau)} + 1\right)e^{t\boldsymbol{\Delta}}\mathbf{R} \tag{24}$$

$$\mathbf{X}(t) = e^{\alpha\tau}e^{-\tau\boldsymbol{\Delta}}\mathbf{X}(t + \tau) + (1 - e^{\alpha\tau})e^{t\boldsymbol{\Delta}}\mathbf{R} \tag{25}$$

## C.3 Stochasticity of the node representation matrix

In Section 3, we mention that if the matrices $e^{t\boldsymbol{\Delta}}$ and $\mathbf{N}$ are both column stochastic for all $t \geq 0$ (Veerman & Lyons, 2020), then $\mathbf{Z}(t) = e^{t\boldsymbol{\Delta}}\mathbf{N}$ is also column stochastic for all $t \geq 0$.

This is easily verified. Let us assume that two matrices $\mathbf{B}$ and $\mathbf{C}$ are column stochastic. They then have nonnegative elements, and they satisfy $\mathbf{1}^\top\mathbf{B} = \mathbf{1}^\top$ and $\mathbf{1}^\top\mathbf{C} = \mathbf{1}^\top$. The matrix $(\mathbf{BC})$ is column stochastic because it has nonnegative elements and satisfies $\mathbf{1}^\top(\mathbf{BC}) = \mathbf{1}^\top\mathbf{BC} = \mathbf{1}^\top\mathbf{C} = \mathbf{1}^\top$.

# D   Consensus

We now give the formulae and constraints for the consensus model (i.e., when $\mathbf{\Delta} = \mathbf{L}^\top$) (DeGroot, 1974).

In this model, the matrix $e^{t\mathbf{\Delta}}$ is row stochastic for all $t \geq 0$. We then constrain both $\mathbf{N} = \mathbf{X}(0)$ and the matrix $\mathbf{M} := \frac{1}{d}\mathbf{1}\mathbf{1}^\top \in \{\frac{1}{d}\}^{n \times d}$ to be row stochastic, this implies $\mathbf{Z}(t)$ row stochastic for all $t \geq 0$. In this case, we have $\forall t \geq 0, e^{t\mathbf{\Delta}}\mathbf{M} = \mathbf{M}$. We then also define $\mathbf{R} := \mathbf{M}$. In the consensus model, $\mathbf{X}(t) = e^{-\alpha t}\mathbf{Z}(t) + (1 - e^{-\alpha t})\mathbf{M}$ is row stochastic for all $t \geq 0$.

A detailed comparison between the diffusion and consensus models is discussed in Veerman & Lyons (2020, Section 6). The consensus model would be appropriate in contexts where each row of $\mathbf{X}(0)$ is a one-hot vector corresponding to the category of the node. However, it might suffer from degenerate cases where some rows of $\mathbf{X}(t)$ do not depend on the same rows at their initial time $t = 0$ due to the properties of the left eigenvectors of $\mathbf{L}$.

# E   Non-symmetric Heat Kernels

We explain how the heat kernel framework in Kondor & Lafferty (2002) is a special case of our encoder (see Section 3) when the kernel function is symmetric and the source term $\mathbf{Q}$ is homogeneous. This connection motivates the study of the column space of the Laplacian matrix that is the foundation of many heat kernel methods (Belkin & Niyogi, 2003; Kondor & Lafferty, 2002).

**Heat kernels (Kondor & Lafferty, 2002).** We first explain how our approach can be seen as a non-symmetric heat kernel when the term $\mathbf{Q}$ is homogeneous. We recall that a function $K : \mathcal{X} \times \mathcal{X} \to \mathbb{R}$ defined on some nonempty set $\mathcal{X}$ is called a *kernel function* if it satisfies Mercer's theorem (Mercer, 1909). In other words, it satisfies $\int_\mathcal{X} \int_\mathcal{X} K(p,q)f(p)f(q)\mathrm{d}p\mathrm{d}q \geq 0$ for every function $f(p)$ of integrable square, or $\sum_{p \in \mathcal{X}} \sum_{q \in \mathcal{X}} K(p,q)f_p f_q \geq 0$ for all sets of real coefficients $\{f_p\}$ in the discrete case, which is the case we are interested in.

Kernel functions are usually defined to be symmetric (i.e., $\forall p,q, K(p,q) = K(q,p)$) to define a Reproducing Kernel Hilbert Space (Smola & Schölkopf, 1998), and the symmetry of the kernel function between pairs of nodes is reasonable when the graph is undirected. However, kernels are not necessarily symmetric (Seely, 1919) and one can define a Reproducing Kernel Banach Space (RKBS) on $\mathcal{X} \supseteq V$ equipped with the $\ell_1$ norm (Song et al., 2011; Zhang et al., 2009) by considering the kernel matrix $\mathbf{K} \in [0,1]^{n \times n}$ defined such that $\mathbf{K} = e^{T\mathbf{\Delta}}$ with $\mathbf{K}_{pq} = K(p,q)$. Let us define $[0,1]^\mathcal{X} := \{f : \mathcal{X} \to [0,1]\}$ the Banach space of functions mapping $\mathcal{X}$ into $[0,1]$. We also define the linear operators $f(\cdot) := \sum_{i=1}^n \alpha_i K(x_i, \cdot)$, $g(\cdot) := \sum_{j=1}^m \beta_j K(\cdot, x'_j)$ and the bilinear form $\langle f, g \rangle := \sum_{i=1}^n \sum_{j=1}^m \alpha_i \beta_j K(x_i, x'_j)$ where $n \in \mathbb{N}, m \in \mathbb{N}, \alpha_i \in \mathbb{R}, \beta_j \in \mathbb{R}, x_1, \ldots, x_n, x'_1, \ldots, x'_m \in \mathcal{X}$ are arbitrary. The resulting RKBS generalizes the heat kernels in Kondor & Lafferty (2002) to directed graphs although it restricts the functions to map into $[0,1]$ instead of $\mathbb{R}$.

When $\mathbf{Q}$ is nonhomogeneous, we have $\mathbf{X}(T) = e^{-\alpha T}\mathbf{Z}(T) + (1 - e^{-\alpha T})\mathbf{M} = \left(e^{-\alpha T}e^{T\mathbf{\Delta}} + (1 - e^{-\alpha T})/n\mathbf{1}\mathbf{1}^\top\right)\mathbf{N}$ since $\frac{1}{n}\mathbf{1}\mathbf{1}^\top\mathbf{N} = \mathbf{M}$ when $\mathbf{N}$ is column stochastic. This leads to the kernel matrix:

$$\mathbf{K} = e^{-\alpha T}e^{T\mathbf{\Delta}} + \frac{1 - e^{-\alpha T}}{n}\mathbf{1}\mathbf{1}^\top \tag{26}$$

As in Laplacian eigenmap approaches for dimensionality reduction (Chung & Yau, 1999), $\mathbf{K}$ can be thought of as an operator on functions defined on nodes of the graph and we obtain the node representation matrix $\mathbf{K}\mathbf{N} \in [0,1]^{n \times d}$. The fact that $\mathbf{K}\mathbf{N}$ is column stochastic allows us to upper bound the $\ell_1$ norm of its columns by 1 (it is equal to 1), and then satisfy the properties of RKBS on $\mathcal{X}$ with the $\ell_1$ norm (Song et al., 2011).

We now explain how our framework falls into the framework of Song et al. (2011); Zhang et al. (2009).

Let us consider the case $V = \mathcal{X}$, which implies $|\mathcal{X}| = n$. We consider the non-symmetric function $K$ as $\mathbf{K}_{pq} = K(p,q)$ where the kernel matrix $\mathbf{K} = e^{-\alpha T}e^{T\mathbf{\Delta}} + (1 - e^{-\alpha T})/n\mathbf{1}\mathbf{1}^\top \in [0,1]^{n \times n}$ is column stochastic.

Since $K(p,q)$ is nonnegative for all $p \in \mathcal{X}$ and $q \in \mathcal{X}$, a sufficient condition to satisfy $\sum_{p \in \mathcal{X}} \sum_{q \in \mathcal{X}} K(p,q)f_p f_q \geq 0$ is to constrain $\{f_p\}_{p \in \mathcal{X}}$ to be a set of nonnegative coefficients. In our experiments, we set $\{f_p\}$ to be a set of nonnegative coefficients that sum to 1 (i.e., column stochastic).

The above explanation did not require the notion of RKBS. We can nonetheless use Proposition 5 of Zhang et al. (2009) which is stated as follows: *If the input space $\mathcal{X}$ is a finite set, then any nontrivial function $K$ on $\mathcal{X} \times \mathcal{X}$ is the reproducing kernel of some RKBS on $\mathcal{X}$.*

In our case, $K$ is nontrivial because $\mathbf{K}$ is full rank. Our kernel matrix (that is full rank, and column stochastic hence with nonnegative elements) naturally satisfies the first three requirements of Song et al. (2011). It also satisfies the relaxation of their fourth requirement in their Section 6. We recall their requirements:

• (A1) for all sequences $\{x_p : p \in \{1, \ldots, n\}\} \subseteq \mathcal{X}$ of pairwise distinct sampling points, the matrix $\mathbf{K} := [K(p, q)] \in \mathbb{R}^{n \times n}$ is non singular.

- (A1) is satisfied because $\mathbf{K}$ is full rank in our case.

• (A2) $K$ is bounded, namely, $|K(s, t)| \le M$ for some positive constant $M$ and all $s, t \in \{1, \ldots, n\}$.

- (A2) is satisfied because $\mathbf{K}$ is column stochastic, so $\forall s, t, |K(s, t)| \le 1$. This is satisfied if $1 \le M$.

• (A3) for all pairwise distinct $p \in \mathcal{X}$, $j \in \mathbb{N}$ and $\mathbf{c}$ having its $\ell_1$-norm finite, $\sum_{j=1}^{\infty} c_j K(x_j, x) = 0$ for all $x \in \mathcal{X}$ implies $\mathbf{c} = \mathbf{0}$.

- (A3) is satisfied because $\forall j, K(x_j, x_j) > 0$ and $\forall j \ne i, K(x_j, x_i) \ge 0$ in our case. Therefore, (A3) can be satisfied only if $\mathbf{c} = \mathbf{0}$.

• the relaxation of (A4) in their Section 6 can be formulated as follows: let us write $\mathbf{K}$ as follows:

$$\mathbf{K} = \begin{pmatrix} \mathbf{K}_{1:(n-1),1:(n-1)} & \mathbf{K}_{1:(n-1),n} \\ \mathbf{K}_{n,1:(n-1)} & \mathbf{K}_{n,n} \end{pmatrix} \tag{27}$$

where $\mathbf{K}_{1:(n-1),1:(n-1)} \in [0, 1]^{(n-1) \times (n-1)}$ and $\mathbf{K}_{1:(n-1),n} \in [0, 1]^{n-1}$ are submatrices of $\mathbf{K}$. The relaxation of (A4) is satisfied if there exists some $\beta_n$ such that: $\|(\mathbf{K}_{1:(n-1),1:(n-1)})^{-1}\mathbf{K}_{1:(n-1),n}\|_{\ell_1} \le \beta_n$ is satisfied. Since $\mathbf{K}_{1:(n-1),n}$ has its values in $[0, 1]$, we have to be able to bound the values of $(\mathbf{K}_{1:(n-1),1:(n-1)})^{-1}$. We can bound them by exploiting the (inverse of the) eigenvalues of $\mathbf{K}$ that depend on $T$ and $\alpha$ (see equation 29) since we know that $\forall r, |\lambda_r + 1| \le 1$.

It is worth noting that this section has proven that it is possible to formulate a RKBS to represent our node similarities. However, the kernel matrix $\mathbf{K}$ is given as input of our algorithm via the adjacency matrix $\mathbf{A}$. One could define some node representation space that would induce the matrix $\mathbf{K}$ by using the theory of RKBS instead of considering that $\mathbf{K}$ is given as input of the algorithm.

## F  Difference between $T$ and $\alpha$

To understand the difference of impact between $T$ and $\alpha$, we need to study the eigenvalues of the kernel matrix described in Appendix E: $\mathbf{K} = e^{-\alpha T}e^{T\mathbf{\Delta}} + (1 - e^{-\alpha T})/n\mathbf{1}\mathbf{1}^\top \in [0, 1]^{n \times n}$.

It is worth noting that $\mathbf{S}$, $\mathbf{\Delta} = \mathbf{L}$ and $e^{T\mathbf{\Delta}}$ all have the same set of right and left eigenvectors. The only difference is their set of eigenvalues. Since $\mathbf{S}$ is column stochastic, it is diagonalizable, its spectral radius is 1 and it has at least one eigenvalue equal to 1 with $\mathbf{1}$ as left eigenvector. The number of eigenvalues of $\mathbf{S}$ that are equal to 1 is the number of *reaches* of the graph (Veerman & Lyons, 2020). A *reach* of a directed graph is a maximal unilaterally connected set (see Veerman & Lyons (2020) for details). By definition of $\mathbf{\Delta} = \mathbf{S} - \mathbf{I}$, $\mathbf{\Delta}$ and $\mathbf{S}$ have the same eigenvectors and those that correspond to the eigenvalue 1 of $\mathbf{S}$, correspond to the eigenvalue 0 of $\mathbf{\Delta}$, and to the eigenvalue $e^0 = 1$ of $e^{t\mathbf{\Delta}}$ for all $t \in \mathbb{R}$. The matrix $e^{t\mathbf{\Delta}}$ is column stochastic for all $t \ge 0$, because its spectral radius is then 1, it has at least one eigenvalue equal to 1 with $\mathbf{1}$ as left eigenvector, and it has the same eigenvectors as the column stochastic matrix $\mathbf{S}$.

Let us note $\mathbf{\Delta} = \mathbf{U}\mathbf{\Lambda}\mathbf{U}^{-1}$ the eigendecomposition of $\mathbf{\Delta}$. The eigendecomposition of $\mathbf{S}$ is $\mathbf{S} = \mathbf{U}(\mathbf{\Lambda} + \mathbf{I})\mathbf{U}^{-1}$, and the eigendecomposition of $e^{t\mathbf{\Delta}}$ is $e^{t\mathbf{\Delta}} = \mathbf{U}e^{t\mathbf{\Lambda}}\mathbf{U}^{-1}$. Let us consider that the first row of $\mathbf{U}^{-1}$ is $\gamma\mathbf{1}^\top$

where $\gamma \neq 0$ is an appropriate factor (i.e., the first row of $\mathbf{U}^{-1}$ is collinear to $\mathbf{1}^\top$), and let us note:

$$\mathbf{\Lambda} = \begin{pmatrix} \lambda_1 & 0 & \dots & 0 & \dots & 0 \\ 0 & \lambda_2 & 0 & \dots & \dots & 0 \\ \vdots & & \ddots & & & \vdots \\ \vdots & & & \ddots & & \vdots \\ 0 & \dots & \dots & 0 & \lambda_{n-1} & 0 \\ 0 & \dots & 0 & \dots & 0 & \lambda_n \end{pmatrix} \tag{28}$$

We know that $\lambda_1 = 0$ since $\mathbf{S}$ is column stochastic. Moreover, both $\frac{1}{n}\mathbf{1}\mathbf{1}^\top$ and $e^{T\mathbf{\Delta}}$ have $\mathbf{1}$ as left eigenvector with corresponding eigenvalue equal to 1, so $\mathbf{K} = e^{-\alpha T}e^{T\mathbf{\Delta}} + (1 - e^{-\alpha T})\frac{1}{n}\mathbf{1}\mathbf{1}^\top$ also has $\mathbf{1}$ as left eigenvector with corresponding eigenvalue equal to 1. The eigendecomposition of $\mathbf{K}$ is then $\mathbf{K} = \mathbf{V}\mathbf{\Phi}\mathbf{V}^{-1}$ where $\mathbf{V} \neq \mathbf{U}$ in general, but the first row of $\mathbf{V}^{-1}$ is collinear to $\mathbf{1}^\top$. The diagonal matrix $\mathbf{\Phi}$ is written:

$$\mathbf{\Phi} = \begin{pmatrix} 1 & 0 & 0 & 0 & \dots & 0 \\ 0 & e^{T(\lambda_2-\alpha)} & 0 & \dots & \dots & 0 \\ 0 & 0 & \ddots & & & \vdots \\ \vdots & & & \ddots & 0 & 0 \\ 0 & \dots & \dots & 0 & e^{T(\lambda_{n-1}-\alpha)} & 0 \\ 0 & \dots & 0 & 0 & 0 & e^{T(\lambda_n-\alpha)} \end{pmatrix} \tag{29}$$

It is worth noting that all the nonzero eigenvalues $\lambda_r$ of $\mathbf{\Delta}$ have negative real part by definition of $\mathbf{S}$ (i.e., since the spectral radius of $\mathbf{S}$ is 1). If $\alpha = 0$, then $\forall r, \lambda_r = 0 \implies e^{T(\lambda_r-\alpha)} = 1$. If $\alpha > 0$ and $T > 0$, then the real part of $\lambda_r - \alpha$ is negative for all $r$, which implies $|e^{T(\lambda_r-\alpha)}| < |e^0| = 1$. If $\alpha > 0$ and $T > 0$, we also have for all $r$, $\lim_{T \to +\infty} |e^{T(\lambda_r-\alpha)}| = 0$ and $\lim_{\alpha \to +\infty} |e^{T(\lambda_r-\alpha)}| = 0$.

From equation 29, the main difference between $T > 0$ and $\alpha > 0$ is that $T$ acts as a multiplicative factor on the eigenvalues inside the exponential, whereas $\alpha > 0$ only has an impact on the real part of the eigenvalue inside the exponential.

## G   Additional Experimental Results

### G.1   Study of the column space of the learned representations

Fig 4 illustrates additional qualitative results from the experiments in Section 6.1 showing cosine values between the singular vectors (ordered by magnitude of their singular values) of $e^{t\mathbf{\Delta}^i}$ and $e^{t\mathbf{\Delta}^i}\mathbf{N}$. To improve visualization, we set all the absolute values lower than 0.3 to 0. One can see that most cosine correlations appear along the diagonal. This shows that the singular vectors of the different matrices are correlated. Although some absolute values are high in the bottom right corner, their corresponding singular values are much smaller compared to those in the top left corner, so the overall importance of their correlation is weaker.

It is worth noting that we use conditional sampling in this experiment so our model has to jointly learn representations that are relevant for both categories.

### G.2   Approximation techniques for larger graphs

In this subsection, we consider a task similar to the one described in Section 6.2. The goal in Section 6.2 was to show that digraphs could be entirely reconstructed with our approach by using low-rank approximations of the Laplacian for small graphs. In other words, if we give some training adjacency matrix $\mathbf{A} \in \{0, 1\}^{n \times n}$ to our noising encoder, then our decoder is able to reconstruct all the elements of $\mathbf{A}$. For simplicity, we did not consider data augmentation techniques in Section 6.2.

We now describe our experimental setup to generate larger graphs of different sizes. As in Section 6.2, we consider the class-conditional digraph generation with two categories. We use the following categories

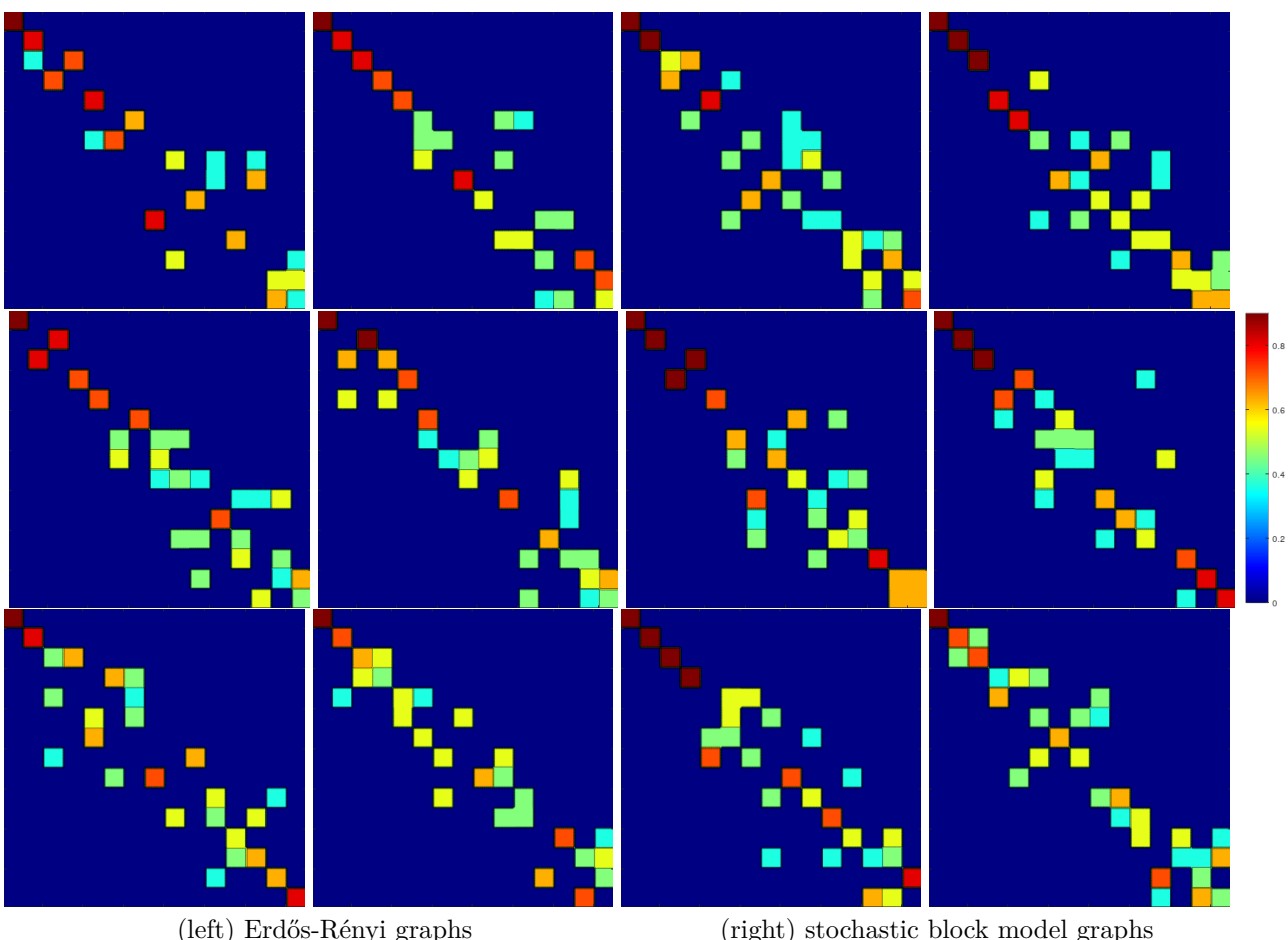

(left) Erdős-Rényi graphs         (right) stochastic block model graphs

Figure 4: Correlations between the left singular vectors of $e^{t\boldsymbol{\Delta}^i}$ and $e^{t\boldsymbol{\Delta}^i}\mathbf{N}$.

Erdős-Rényi with $p = 0.4$, and a stochastic block model with 5 blocks and the following transition matrix between the blocks:

$$\boldsymbol{\Pi} = \begin{pmatrix} 0.9 & 0.2 & 0.4 & 0.2 & 0.4 \\ 0.3 & 0.9 & 0.15 & 0.5 & 0.45 \\ 0.4 & 0 & 0.95 & 0.05 & 0.4 \\ 0 & 0.3 & 0.4 & 0.75 & 0.45 \\ 0.1 & 0.4 & 0.4 & 0.15 & 0.7 \end{pmatrix} \tag{30}$$

All the graphs contain $n_i$ nodes where $n_i \in \{180, 181, \dots, 200\}$. The first block contains 40 nodes, the second block contains 20 nodes, the third and fourth blocks contain 35 nodes each, and the last block contains from 50 to 70 nodes. We set $\mu = 0.4$, $\rho = 1/n_i$ and $s = 50$ for the rank-$s$ approximation of $e^{T\boldsymbol{\Delta}^i} \in [0,1]^{n_i \times n_i}$ in this experiment.

At inference time, we sample digraphs that contain $n_i$ nodes where $n_i \in \{180, 181, \dots, 200\}$. Quantitative results are reported in Table 2 and Table 3. Once again, DGDK outperforms GRAN and SwinGNN in evaluation metrics that take into account global properties of the graph. Data augmentation slightly improves performance.

In both Table 2 and Table 3., we report the MMD scores for different values of the variance parameter in the RBF kernel: $\sigma^2 = 100, 10$ and $1$. We recall that we use $\sigma^2 = 100$ in Table 1. We call *Discrete DGDK* the sampling strategy described in Algorithm 1 where we sample discrete adjacency matrices $\mathbf{A} \in \{0,1\}^{n \times n}$ as described in line 2 of Algorithm 1. *Continuous DGDK* corresponds to the same sampling strategy except

Table 2: Squared MMD distances over 5 random initializations (average ± standard deviation) for the Erdős-Rényi distribution ($p = 0.4$).

| | MMD metric | Degree | Clustering | Spectrum |
|---|---|---|---|---|
| $\sigma^2 = 100$ | Sampling from flat Dirichlet distribution | 0.00091 ± 0.00043 | 0.0079 ± 0.0026 | 0.00193 ± 0.0004 |
| | Continuous DGDK (with data augmentation) | 0.00074 ± 0.00004 | 0.0068 ± 0.0005 | **0.00085 ± 0.0001** |
| | Discrete DGDK (with data augmentation) | **0.00073 ± 0.00002** | **0.0067 ± 0.0006** | **0.00085 ± 0.0001** |
| | Discrete DGDK (without data augmentation) | 0.00082 ± 0.00003 | 0.0069 ± 0.0004 | 0.00092 ± 0.0003 |
| | SwinGNN | 0.00091 ± 0.00084 | 0.0091 ± 0.0012 | 0.00110 ± 0.0008 |
| | GRAN | 0.00118 ± 0.00124 | 0.0138 ± 0.0035 | 0.00143 ± 0.0013 |
| $\sigma^2 = 10$ | Sampling from flat Dirichlet distribution | 0.0093 ± 0.0065 | 0.077 ± 0.0298 | 0.0202 ± 0.003 |
| | Continuous DGDK (with data augmentation) | 0.0074 ± 0.0005 | 0.067 ± 0.0049 | 0.0085 ± 0.001 |
| | Discrete DGDK (with data augmentation) | 0.0072 ± 0.0002 | 0.065 ± 0.0009 | 0.0085 ± 0.001 |
| | Discrete DGDK (without data augmentation) | 0.0083 ± 0.0004 | 0.071 ± 0.0014 | 0.0123 ± 0.002 |
| | SwinGNN | 0.0090 ± 0.0035 | 0.079 ± 0.0021 | 0.0113 ± 0.006 |
| | GRAN | 0.0121 ± 0.0103 | 0.143 ± 0.0241 | 0.0149 ± 0.012 |
| $\sigma^2 = 1$ | Sampling from flat Dirichlet distribution | 0.103 ± 0.098 | 0.57 ± 0.14 | 0.184 ± 0.0340 |
| | Continuous DGDK (with data augmentation) | 0.076 ± 0.009 | 0.57 ± 0.08 | 0.084 ± 0.0122 |
| | Discrete DGDK (with data augmentation) | 0.071 ± 0.002 | 0.54 ± 0.02 | 0.083 ± 0.0092 |
| | Discrete DGDK (without data augmentation) | 0.084 ± 0.004 | 0.60 ± 0.02 | 0.103 ± 0.0104 |
| | SwinGNN | 0.102 ± 0.064 | 0.64 ± 0.10 | 0.101 ± 0.0126 |
| | GRAN | 0.142 ± 0.105 | 1.39 ± 0.22 | 0.145 ± 0.1156 |

Table 3: Squared MMD distances over 5 random initializations (average ± standard deviation) for the stochastic block model (5 blocks).

| | MMD metric | Degree | Clustering | Spectrum |
|---|---|---|---|---|
| $\sigma^2 = 100$ | Sampling from flat Dirichlet distribution | 0.00031 ± 0.0002 | 0.0069 ± 0.0026 | 0.00232 ± 0.0004 |
| | Continuous DGDK (with data augmentation) | 0.00017 ± 0.0001 | 0.0057 ± 0.0002 | 0.00041 ± 0.0003 |
| | Discrete DGDK (with data augmentation) | **0.00015 ± 0.0001** | **0.0039 ± 0.0023** | **0.00038 ± 0.0003** |
| | Discrete DGDK (without data augmentation) | 0.00031 ± 0.0001 | 0.0046 ± 0.0025 | 0.00041 ± 0.0004 |
| | SwinGNN | 0.00046 ± 0.0009 | 0.0245 ± 0.0094 | 0.00831 ± 0.0103 |
| | GRAN | 0.00053 ± 0.0008 | 0.0654 ± 0.0057 | 0.02472 ± 0.0144 |
| $\sigma^2 = 10$ | Sampling from flat Dirichlet distribution | 0.0021 ± 0.001 | 0.067 ± 0.020 | 0.0187 ± 0.002 |
| | Continuous DGDK (with data augmentation) | 0.0018 ± 0.001 | 0.059 ± 0.006 | 0.0040 ± 0.003 |
| | Discrete DGDK (with data augmentation) | 0.0015 ± 0.001 | 0.056 ± 0.001 | 0.0039 ± 0.003 |
| | Discrete DGDK (without data augmentation) | 0.0027 ± 0.002 | 0.059 ± 0.004 | 0.0043 ± 0.004 |
| | SwinGNN | 0.0041 ± 0.003 | 0.264 ± 0.062 | 0.0793 ± 0.089 |
| | GRAN | 0.0056 ± 0.004 | 0.664 ± 0.091 | 0.2629 ± 0.124 |
| $\sigma^2 = 1$ | Sampling from flat Dirichlet distribution | 0.0255 ± 0.0133 | 0.59112 ± 0.342 | 0.172 ± 0.031 |
| | Continuous DGDK (with data augmentation) | 0.0186 ± 0.0132 | 0.58531 ± 0.162 | 0.041 ± 0.036 |
| | Discrete DGDK (with data augmentation) | 0.0154 ± 0.0093 | 0.48165 ± 0.012 | 0.038 ± 0.032 |
| | Discrete DGDK (without data augmentation) | 0.0331 ± 0.0103 | 0.56420 ± 0.025 | 0.048 ± 0.042 |
| | SwinGNN | 0.0424 ± 0.0251 | 0.68073 ± 0.351 | 0.388 ± 0.724 |
| | GRAN | 0.0553 ± 0.0421 | 1.56743 ± 0.931 | 1.253 ± 1.123 |

that we sample non-diagonal elements of $\mathbf{A}$ uniformly in the continuous interval $[0, 1]$ instead of $\{0, 1\}$, while keeping the constraint $\forall i, \mathbf{A}_{ii} = 1$. The two methods are competitive with baselines.

On the other hand, we also report scores when we sample each column of the input directly from a flat Dirichlet distribution. This strategy does not exploit the learned matrix $\mathbf{N}$ and is outperformed by our other sampling strategies although it still outperforms the GRAN deadline.

## G.3    Data augmentation by using permutations of Laplacian matrices

In this subsection, we study the impact of the data augmentation technique adding adjacency matrices of isomorphic digraphs as explained in Section 4.2. Figure 5 illustrates the loss value of equation 7 as a function of the number of iterations with and without this data augmentation technique. In this setup, we use $\gamma = 100$.

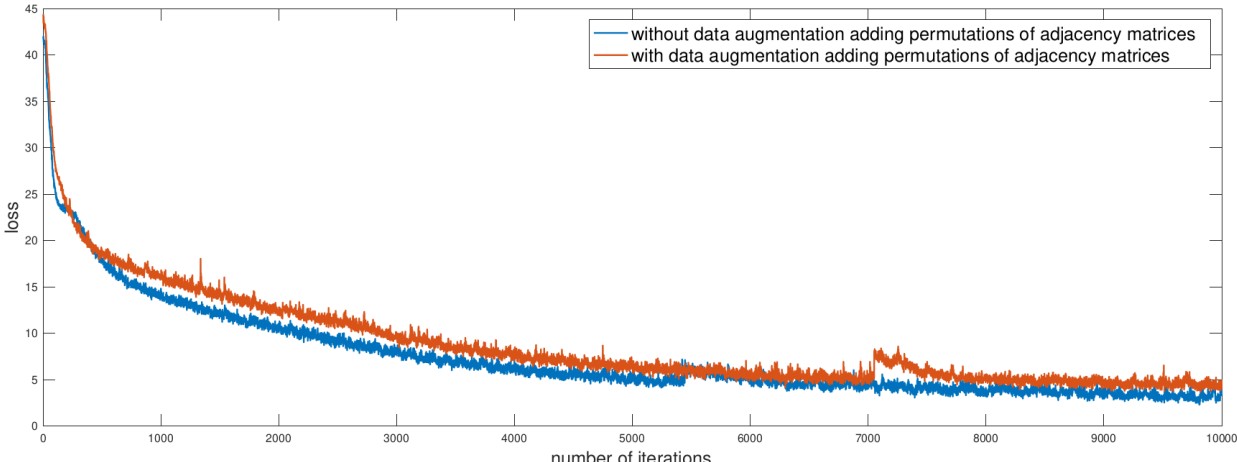

Figure 5: Loss values obtained when optimizing equation 7 with or without data augmentation by adding permutations of training adjacency matrices.

We use the experimental setup described in Section 6.2 with 3,000 training graphs per category, 10 graphs per category per batch. Each epoch corresponds to 300 iterations. Both loss curves follow similar patterns, although the one corresponding to data augmentation has a slightly higher loss value.

One reason of the low impact of this data augmentation technique is that the matrix $\mathbf{N}$ is jointly learned with nonlinear decoders $\varphi$ and $\psi$ that are robust to this kind of transformation.

### G.4 Additional experiment: multimodal categories

In Section 6.2, the categories are unimodal and it is assumed that they can be conditioned over at test time. We now show that if the distribution of the training set is multimodal and the modes are not given to the model, our model is able to sample graphs from the different modes.

We consider the following training set containing four modes of same size:

(1) a single block stochastic model with probability $p = 0.28$,

(2) two (disjoint and connected) components of same size with $p = 0.48$ each,

(3) three components of same size with $p = 0.78$ each,

(4) and four components of same size with $p = 0.97$ each.

This corresponds to an average edge ratio of $\mu \approx 0.24$.

Fig. 7 illustrates some graphs generated with Algorithm 1 when $\gamma = 1$ and $\alpha = 1$, they follow the training distribution. We emphasize that blocks are disjoints only in this subsection for visualization purpose, not in the previous subsections where some edges connect different blocks.

**Impact of $\gamma$.** The regularization parameter $\gamma$ in equation 7 acts as a tradeoff between learning the edge decoder and the node decoder. If $\gamma = 0$, then the node decoder is not trained. Our goal is to predict the adjacency matrix from a random matrix in $[0, 1]^{n \times d}$ given as input of the decoders. Only the edge decoder is useful for this task. We found that the loss function does not converge during training when $\gamma = 0$. This suggests that the data augmentation technique that adds edge perturbation to the adjacency matrix is important, at least for this experiment. A positive value of $\gamma$ is necessary for the loss to decrease during training and learn meaningful decoders. This implies that the node decoder learns the global structure of the graph by reconstructing node representations that depend on the ground truth Laplacian. This is beneficial to the edge decoder as both decoders share a common backbone.

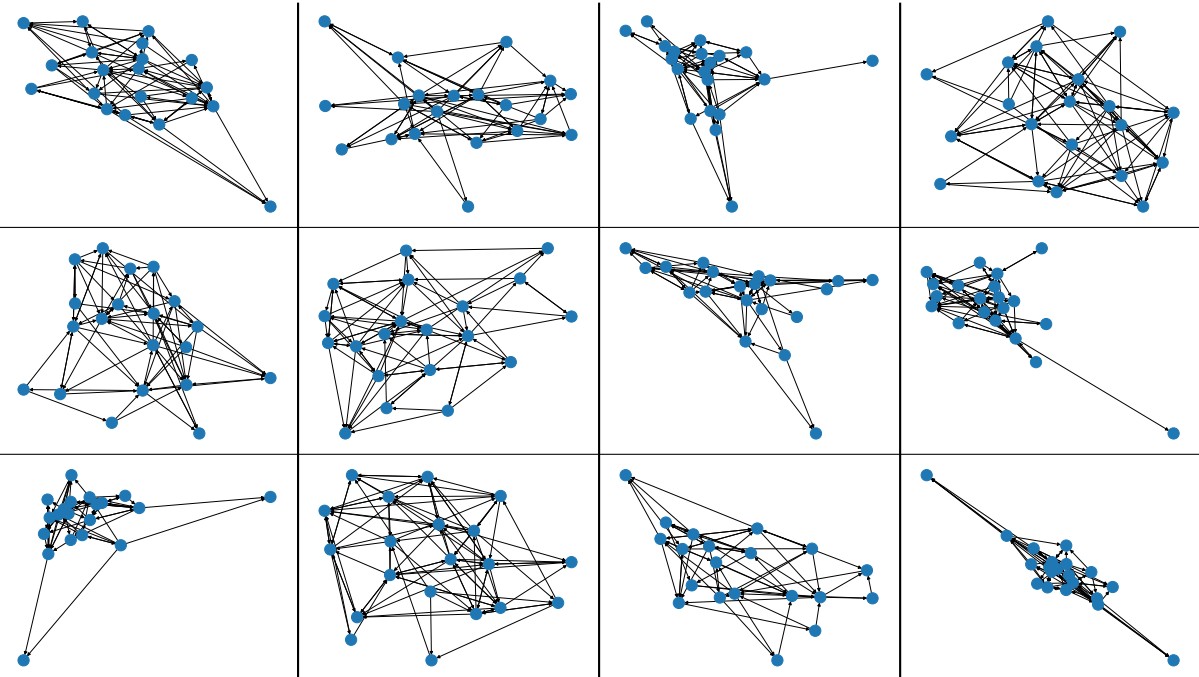

Figure 6: Digraphs generated when $\alpha = 0$.

**Impact of $\alpha$.** Using $\alpha = 0$ results in generated samples with only one block (see Fig. 6). As $\alpha$ increases, the number of components per sample increases (see Fig. 7). This is because the sampled adjacency matrices in Algorithm 1 generated with a Bernoulli distribution correspond to one component. As $\alpha$ increases, the information of the sampled adjacency matrix gets partially lost and the decoder is able to reproduce graphs similar to the training distribution, and it samples graphs uniformly from all the modes. We provide some ablation study on the impact of the noise diffusivity rate hyperparameter $\alpha \geq 0$.

When $\alpha = 0$, more than 97% of the generated graphs contain only one block. Some generated graphs with $\alpha = 0$ are illustrated in Figure 6.

When $\alpha = 1$ (i.e., $1 - e^{-\alpha T} \approx 0.63$), 19% of the generated graphs contain a single block, 26% contain 2 blocks, 30% contain 3 blocks, 20% contain 4 blocks, 2% contain 5 blocks, 2% contain 6 blocks, 1% contain 7 blocks. The distribution is similar to the training set that is uniformly distributed (i.e., 25% for each mode). Some generated graphs when $\alpha = 1$ are illustrated in Figure 7.

When $\alpha = 2.3$ (i.e., $1 - e^{-\alpha T} \approx 0.9$), 17% of the generated graphs contain a single block, 14% contain 2 blocks, 20% contain 3 blocks, 20% contain 4 blocks, 8% contain 5 blocks, 5% contain 6 blocks, 8% contain 7 blocks, and the remaining 8% contains up to 15 blocks. Some graphs generated when $\alpha = 2.3$ are illustrated in Figure 8. Many generated graphs contain nodes that are isolated.

We report in Table 4 the MMD scores that compare the training set with the generated graphs for different values of $\alpha$. The performance shows that $\alpha$ has to be chosen carefully so that noise is introduced, but not in excess. It is worth noting that GRAN is not appropriate in this experiment since it has to be given the maximum number of nodes in the graph. Our model is given a number of nodes and has to generate one or multiple (disconnected) graphs that follow the training distribution.

Table 4: Squared MMD distances for the experiments on the multimodal dataset in Section G.4 for different values of $\alpha$.

| Dataset | Multimodal dataset | | |
|---|---|---|---|
| MMD metric | Degree | Clustering | Spectrum |
| $\alpha = 0$ | $2.6 \times 10^{-3}$ | $1.7 \times 10^{-3}$ | $3.3 \times 10^{-4}$ |
| $\alpha = 1$ | $0.9 \times 10^{-3}$ | $6.3 \times 10^{-4}$ | $2.3 \times 10^{-4}$ |
| $\alpha = 2.3$ | $1.6 \times 10^{-3}$ | $1.0 \times 10^{-3}$ | $5.9 \times 10^{-4}$ |

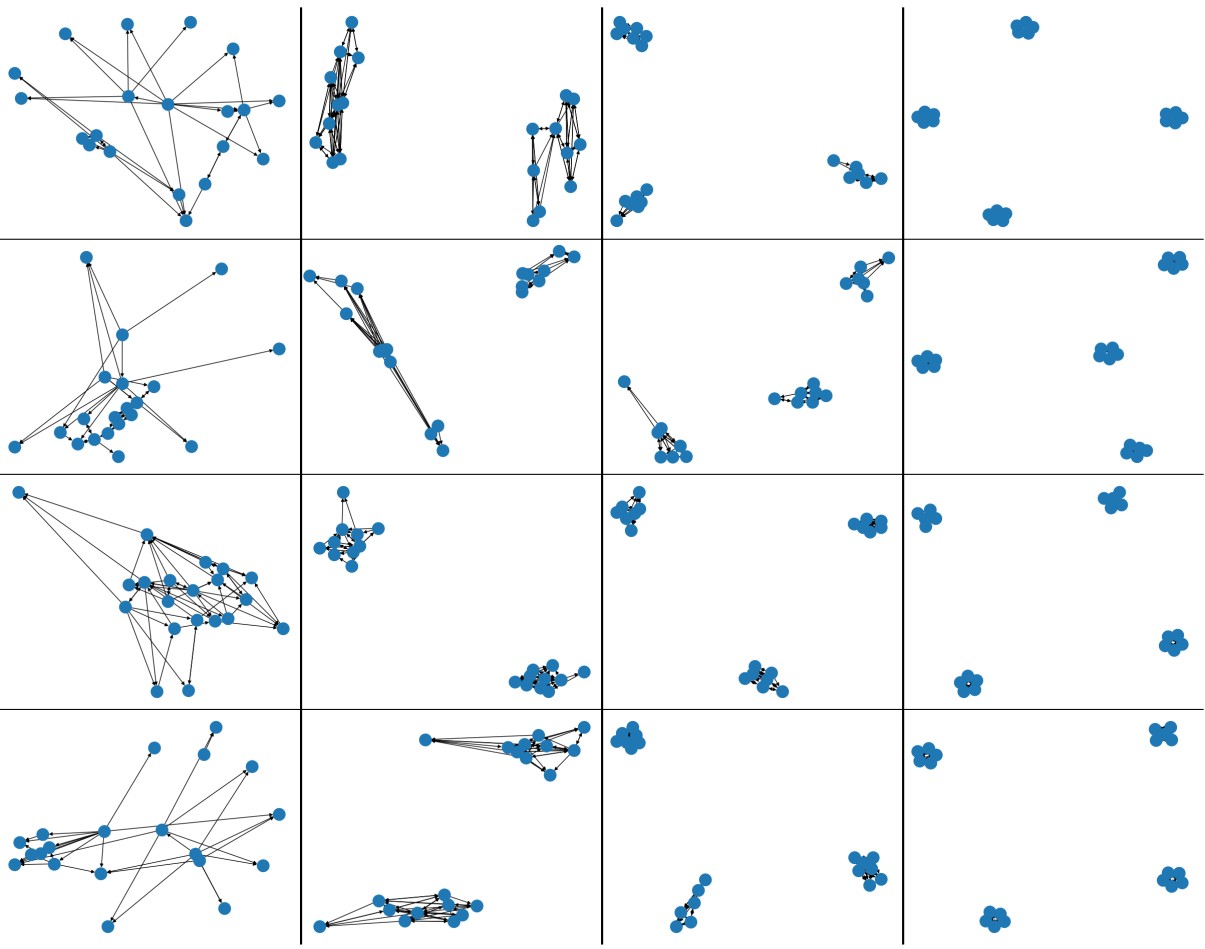

Figure 7: Digraphs generated when $\alpha = 1.0$.

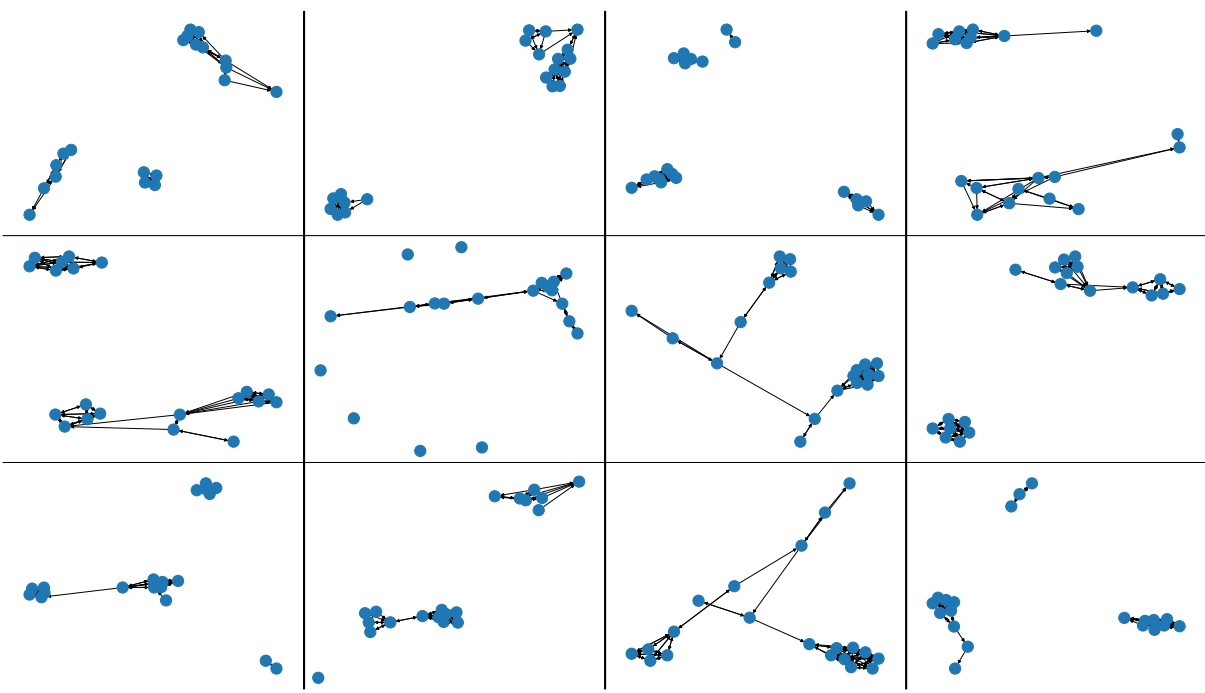

Figure 8: Digraphs generated when $\alpha = 2.3$.

