# OpenReview forum: "Directed Graph Generation with Heat Kernels"
_TMLR — Accepted by TMLR_

### Review · Reviewer_i1zX · 2024-10-22

**Summary Of Contributions:**

The authors propose a method for one-shot directed graph generation with denoising auto encoder-based generative model using the heat equation with the graph Laplacian. The input graph is perturbed using augmentation techniques which is then modeled with an encoder which generates noise based on the heat equation, thus not relying on a neural network to add noise. The output of the encoder is then used to reconstruct the original input with diffusion leveraging random walk Laplacian. The framework is evaluated on synthethic datasets.

**Audience:**

Yes

**Broader Impact Concerns:**

No concerns.

**Claims And Evidence:**

Yes

**Requested Changes:**

- The paper would be significantly strengthened by adding evaluation on at least one real-world dataset

**Strengths And Weaknesses:**

Strengths:
- The method focuses on directed graph generation, which is an important problem, particularly as the majority of the methods focused solely on undirected graphs
- The proposed approach is original and novel, particularly the application of heat equation for modeling noise
- The empirical setup & evaluation is sound and show the benefit of the proposed method

Weaknesses:
- Perhaps the biggest weakness of the paper is that the evaluation is performed solely on the digraphs and synthetic datasets, thus not providing sufficient evidence to the community whether it could be applied in practice to real-world datasets. For example on gene network inference [1], where having direction is particularly important.
- In general I have found sections 2 and 3 slightly hard to read, with often the mathematical notation not backed by motivation. It would be worth spending more time on explaining the reasoning behind the methods and move some of the items to the appendix.


[1] Marbach et al. Wisdom of crowds for robust gene network inference. Nature Methods, 9(8):796–804, 2012. ISSN 1548-7091.

---

> ### Author Response · Authors · 2024-11-07
> **Response to Reviewer i1zX**
>
> $\bullet$ We agree that the lack of real-world dataset is a weakness of our paper. We did not find an appropriate real-world dataset to try our approach. Even the dataset by Marbach et al [1] contains 3 different graphs, each for a different type of gene. Our goal is to work with multiple graphs that follow the same kind of distribution. On the other hand, we would have to work with subgraphs of [1] to generate other subgraphs, which is different from the task we are interested in.
>
> We would like to emphasize that we used larger graphs in Appendix G.2.
>
> $\bullet$ Following your comments, we have improved Sections 2 and 3 to give more intuitions to the reader. We added a paragraph about heat kernels in Section 2, and an interpretation of the heat equation, which is basically a continuous way to perform message passing. Our nonhomogeneous term perturbs (deterministically) our node representations to tend to a target non-informative matrix.
>
> We also tried to improve the readability of Section 3.

---

> > ### Comment · Reviewer_i1zX · 2024-11-26
> >
> > - I understand that the approach may not currently be suitable to the real-world datasets and I appreciate the clarification on it. This is something which I would mention in the section on limitations.
> >
> >
> > - The incorporated changes make it easier to follow the manuscript and understand the key contributions.

---

> > > ### Author Response · Authors · 2024-11-27
> > > **Thank you for the suggestion**
> > >
> > > Thank you for the suggestion! We have updated the limitation paragraph accordingly.

---

### Review · Reviewer_e7P4 · 2024-10-23

**Summary Of Contributions:**

This work presents a novel approach for generating graphs in a physically inspired way. Their physical model of choice is the driven heat equation, and they develop a technique that can generate a general graph (either directed or undirected) that is similar to the training set in one shot.

**Audience:**

Yes

**Claims And Evidence:**

Yes

**Requested Changes:**

Overall the paper is well written and self contained. I have a few comments:

- It would be nice if the author could comment on the scalability of this technique, or approximations that might be applicable. Additionally, it would be nice to have a runtime table in the text
- One of the core properties of directed graphs are the presence of strongly connected components -- It would be nice to see an analysis of whether DGDK preserves these clusters. I understand that the MMD is low, but it is possible to have a low MMD *and* destroy the community structure.
- Please comment on the importance of your link predictor? Does using a symmetric edge-encoding appreciably change the efficacy of your decoding procedure?
- Could you present an ablation on $\gamma$?
- The noise parameter, $\rho$, was set to $1/n_i$. How does the handle higher noise regimes? Or lower ones?
- Would a directed node encoder (e.g. DirGNN or MagNET) improve this model?

**Strengths And Weaknesses:**

**Strengths**
- The paper is clearly written and easy to follow. I was able to reimplement their work with little effort
- DGDK is clearly a novel and interesting method
- The work is extremely well motivated

**Weaknesses**
- Scalability. It appears that these experiments only used 21 vertices and a handful of edges.
- Some of the notation is a bit hard to follow in spots. I might have missed it but it was challenging to realize that $M$ is a constant matrix, for example.

---

> ### Author Response · Authors · 2024-11-07
> **Response to Reviewer e7P4**
>
> Thank you for your positive review. The main strength of our approach is its simplicity.
>
> About weaknesses:
>
> $\bullet$ **Scalability.**
>
> In the main paper, we indeed consider experiments with graphs with only 21 vertices. However, we consider experiments with larger graphs (up to 200 nodes) in Appendix G.2. We follow the same protocol evaluation as in Section 6.2 and obtain results that follow the same trend. We report the running times in Appendix B. In particular, our training algorithm takes about 1 hour for 10,000 iterations.
> The main bottleneck of our approach is the memory to store the training graphs and the neural network. Since we ran our experiments on a single GPU with 24GB of VRAM, we did not extend to larger graphs. However, this is easily feasible on clusters with larger memory.
>
> $\bullet$ Following your comments, we have improved Sections 2 and 3 to give more intuitions to the reader, and we have added that $\textbf{M} := \frac{1}{n} \mathbf{1} \mathbf{1}^{\top} \in \{ \frac{1}{n}\}^{n \times d}$ is a constant matrix.
>
>
> $\bullet$ **Discussion about scalability**
>
> We have added a discussion about scalability in Section 7.1, which also includes training times. Those times were already previously reported in the appendix but we have moved them to the main paper. Since our framework uses efficient closed-form solutions, the main limitation is memory to store the kernel matrix $K$ which can be preprocessed online. This kernel matrix grows quadratically in the number of nodes, but our proposed low-rank approximation of the matrix allows us to keep the required memory linear in the number of nodes.
>
> $\bullet$ **Experiments about strongly connected components.**
>
> We have an experiment in Appendix G.4 that considers four different modes and is similar to your request. The four modes are as follows:
> (1) we consider graphs with a single connected component and connectivity/transition probability $p=0.28$,
> (2) two (disjoint) connected components with blocks of same size with $p = 0.48$  each,
> (3) three components of same size with $p=0.78$ each,
> (4) four components of same size with $p=0.97$.
>
> Our experiment shows that DGDK preserves is able to generate samples from all four modes and respect the original transition probability (see Figure 7). We also provide an ablation on both $\gamma$ and $\alpha$ in Appendix G.4. We also tried lower values of $\rho$, the results were relatively similar and did not seem to affect the quantitative results much. We haven't tried higher values
>
> $\bullet$ Technically, a directed node encoder could also be used since our encoder and decoder could be used independently. We assume that the results could be improved with another neural network. The main advantage of our encoder is that it is efficient and obtained in closed form, whereas an encoder based on a neural network would have to be trained.
>
> $\bullet$ **Importance of the link predictor**
>
> We do not understand what you mean by symmetric edge-encoding? We use a nonsymmetric edge-encoder since we work with digraphs.  Our link/edge predictor predicts the edges that are returned by our denoising decoder.

---

### Review · Reviewer_VUaS · 2024-10-23

**Summary Of Contributions:**

This paper presents a denoising auto-encoder-based generative model for directed graphs. The key idea is to exploit a heat-diffusion-based encoder that maps the “noisy” adjacency matrix to noisy node representations. The directed graphs are then reconstructed by a decoder. Experiments on small graph datasets show promising results.

**Audience:**

Yes

**Claims And Evidence:**

No

**Requested Changes:**

The requested changes are in my comments (cons) in Strengths And Weaknesses.

**Strengths And Weaknesses:**

Pros:

- The idea of using heat diffusion to encode the graph is interesting in the context of graph generation. There are a few benefits with such a design: 1) one can explicitly control the diffusion speed via the designed diffusion equation; 2) the closed-form solution to the heat diffusion allows fast computation.

- The connection with heat kernel methods is nice, as the proposed encoder incorporates them as special cases. This part is currently presented in the Appendix. It would be great to move some parts in the main paper.

- The experiments on small-scale graph datasets show promising results compared to the auto-regressive models.

Cons and Suggestions:

- Although the proposed method is closely related to denoising auto-encoder, it lacks a principled probabilistic model. In particular, during generation, one has to first sample an adjacency matrix from the Erdős–Rényi model and then inject noise into it by randomly corrupting the adjacency matrix. This part is somewhat ad-hoc since the training distribution of adjacency matrices is typically very different from the Erdős–Rényi model, where edges are independent. The authors argue that this procedure is similar to using a prior in a variational auto-encoder (VAE). However, the proposed model does not have a probabilistic formulation like VAE, e.g., there is no ELBO to constrain the discrepancy between the prior and the encoder distribution. I hope the authors could discuss this part and provide more convincing arguments.

- Reproducing Kernel Banach Spaces (RKBS) has been mentioned in the abstract as a way to generalize a special class of exponential kernels to the non-symmetric case. However, the authors do not explicitly discuss it in the full paper. Some discussion about the classic Heat kernel (Kondor & Lafferty, 2002) involves RKBS and is only provided in the appendix. It is unclear what the exact technical contributions the authors have made regarding this part. It would be great to highlight the exact technical contribution w.r.t. RKBS in the main paper.

- It has been mentioned in Proposition 1 and some other places that Q is the so-called heat source term that introduces noise in heat diffusion. However, in the proposed definition of Q in Proposition 1, no variables are random. The description is confusing.

- In principle, one can add any “noise” to the adjacency matrix within this framework. The authors mention that they followed Veličković et al. (2019) to set the corrupting probability. However, the reference is about unsupervised graph representation learning, whereas this paper is about graph generation. It is unclear if the setup can be directly transferred. It would be great to provide an ablation study on the effect of different types of noise and the noise level.

- Learning a set of input node representations N shared by all graphs makes sense for the purpose of generating adjacency matrices. However, it seems hard to utilize it if one wants to additionally generate node attributes that differ from graph to graph, e.g., atom types in molecule graphs. I hope the authors could discuss the generalization of this framework to attributed graphs.

- The experiments need more baselines. The proposed method is currently only compared with the auto-regressive models. There are quite a few latest methods, including discrete diffusion [1] and continuous diffusion [2]. In particular, for continuous diffusion [2], the denoiser treats the adjacency matrix as an image. Therefore, it can handle both directed and undirected graphs in a unified manner. It would be great to add additional comparisons.

- The experiments need some real-world datasets. Currently, only synthetic datasets are used, and the graph sizes are somewhat limited, e.g., the number of nodes in Section 6.1 is 15, and the number of nodes in Section 6.2 is 21. It would be great to include experiments on larger directed graphs from real-world problems. This would help strengthen the motivation of studying directed graph generation.

Minor:

- In proposition 1, it would be better to restate the goal explicitly so that the proposition becomes self-contained.

- In the 1st paragraph of Section 3, “To this end, we consider that Eq. equation 2 is the output of our heat diffusion encoder…” should be “To this end, we consider that Eq. 2 is the output of our heat diffusion encoder…”

[1] Vignac, C., Krawczuk, I., Siraudin, A., Wang, B., Cevher, V. and Frossard, P., 2023 DiGress: Discrete Denoising diffusion for graph generation. In International Conference on Learning Representations.

[2] Yan, Q., Liang, Z., Song, Y., Liao, R. and Wang, L., 2024. Swingnn: Rethinking permutation invariance in diffusion models for graph generation. Transactions on Machine Learning Research.

---

> ### Author Response · Authors · 2024-11-07
> **Response to Reviewer VUaS**
>
> Thank you for your review.
>
> $\bullet$ **Lack of probabilistic model and connection with VAEs.**
>
> We agree that we do not use a probabilistic model. Our approach is a standard denoising autoencoder whose encoder is calculated in closed form.
> First, we would like to clarify that we do not exactly sample an adjacency matrix from the Erdős–Rényi model. We sample a random matrix that follows global statistics similar to the training set (although we indeed sample from the Erdős–Rényi model in this case) and we apply the same kind of perturbation/data augmentation that is performed on the training set.
> Second, our main motivation to compare our approach to VAEs was to justify the fact that our encoder maps to a distribution similar to a flat Dirichlet distribution. Therefore, at inference time, each column of our input can be sampled from a flat Dirichlet distribution. We compare this flat Dirichlet sampling strategy to our discrete and continuous sampling strategies that sample discrete and continuous adjacency matrices, respectively and are given as input of an encoder, and its output is given to a decoder. The quantitative results can be found in Tables 2 and 3 in the appendix. Although the flat Dirichlet sampling strategy is competitive, the other sampling strategies perform better. Our proposed sampling strategy is standard in the denoising autoencoder literature. We have added a sentence in the paragraph that mentions VAEs in Section 5 to justify this connection.
>
> $\bullet$ **Reproducing Kernel Banach Spaces (RKBS).**
>
> Thank you for the suggestion. We have added a discussion in Section 7.2 about our contribution and motivation for RKBSs. In particular, heat kernels are used in  Belkin and Niyogi (2003) to create low-dimensional representations of the nodes, and in  Kondor and Lafferty (2002) to perform classification by using the kernel matrix as a similarity measure between nodes. In both cases, their graphs are undirected and their kernel matrix is symmetric, which allows them to exploit the connection with reproducing kernel Hilbert spaces (RKHS).
>
> In our case, our kernel matrix is non-symmetric. Therefore, we cannot use RKHS which are based on (symmetric) inner products. Instead, we propose to show that our non-symmetric matrix $K$ still corresponds to a non-symmetric kernel matrix induced by a RKBS. This motivates our low-rank approximation, inspired by kernel methods, to reduce the memory that grows only linearly in the number of nodes instead of quadratically.  Moreover, we propose to add a nonhomogeneous term to introduce noise in our kernel matrix.

---

> > ### Author Response · Authors · 2024-11-07
> > **Response to Reviewer VUaS**
> >
> > $\bullet$ **No random variable in our proposed heat source term.**
> >
> >  It is true that our heat source term that introduces noise that is deterministic and not a Wiener process like in most diffusion methods in machine learning diffusion approaches.
> > We would like to cite Chapter 4.1 of ref [A] to answer this concern: "The inclusion of random effects in differential equations leads to two distinct classes of equations, for which the solution processes have differentiable and non differentiable sample paths, respectively. They require fundamentally different methods of analysis. The first, and simpler, class arises when an ordinary differential equation has random coefficients, a random initial value or is forced by a fairly regular stochastic process, or when some combination of these holds. The equations are called random differential equations and are solved sample path by sample path as ordinary differential equations. The sample paths of the solution processes are then at least differentiable functions."
> >
> > Our approach corresponds to this class of method that contains random coefficients that modify the input of our encoder before being decoded to return the ground truth objects. Our encoder can be efficiently calculated in closed form. On the other hand, Wiener processes may be more challenging to scale when combined with the Laplacian of the graph.
> >
> > [A] Peter Eris Kloeden, Eckhard Platen, and Henri Schurz. Numerical solution of SDE through computer experiments. Springer Science and Business Media, 2012
> >
> > $\bullet$ **About the choice of the hyperparameter $\rho$ for edge perturbation.** The lower $\rho \in [0, 1]$, the more similar the modified adjacency matrix is to the ground truth adjacency matrix during training. We arbitrarily defined $\rho = 1/n$ like in Veličković et al. (2019), but other values could be chosen. We tried lower values and they didn't have much impact on the results, so we picked this value arbitrarily to match with Veličković et al. (2019) although their task is different. We provide in Appendix G.2 different sampling strategies resulting from different types of noise.
> >
> >
> > $\bullet$  **If one wants to generate categorical node attributes,** then the consensus model (which considers that $N$ is row-stochastic instead of column-stochastic) can be used instead of our approach. This corresponds to defining $\Delta = L^{\top}$ instead of $\Delta = L$. We provide all the necessary equations in Appendix D for this context.
> >
> > $\bullet$ **Including ref [2] as a baseline.**
> >
> > Thank you for the suggestion, we didn't have time to include it in the current version but we will include a comparison to ref [2] in the next version. Please note that our Appendix G.2 contains experiments with larger graphs.

---

### Author Response · Authors · 2024-11-07
**General response**

We thank the reviewers for their positive feedback in general. We recall that we propose an approach based on heat kernels to generate directed graphs (or digraphs) in an efficient manner. Our approach can be seen as a denoising autoencoder where the encoder is calculated in closed form by solving the heat equation (for digraphs) and the noise is introduced via a nonhomogeneous term that makes our solution tend to a matrix with all its elements equal.

We would like to emphasize some theoretical contributions. One contribution in Appendix E is the interpretation of our framework via the lens of Reproducing Kernel Banach Spaces (RKBS), which is more general than standard heat kernel approach that use Reproducing Kernel Hilbert Spaces (RKHS) that use (symmetric) inner products between nodes. Another contribution is the study of the spectrum of our kernel matrix to improve the interpretation of our method in Appendix F.

As requested by the reviewers, we have improved Section 2 and 3, and added discussions about scalability and RKBS in Section 7.

---

### Decision · Action_Editor_4E7y · 2024-12-02

**Recommendation:** Accept with minor revision

**Comment:**

Overall, all three reviewers find the work interesting and support acceptance of the paper.

The paper was improved during the discussion paper, and most of the reviewers' concerns were addressed. Reviewer VUaS asked for comparisons to ref. [2], and the authors agreed to include it in the next version of the paper. I encourage the authors to include those comparisons in the final version.

**Audience:**

The paper is of interest to researchers working on generative modeling, and especially generative models of graphs.

**Claims And Evidence:**

This paper presents a generative model for directed graphs, based on a denoising auto-encoder. It uses a heat-diffusion-based encoder to produced noisy node representations, and a decoder that reconstructs the directed graphs from those representations. Experiments are carried out on small graph datasets, showing promising results (the paper also includes experiments on larger graphs in the Appendix, and discusses scalability).

Overall, all three reviewers found that the paper presents interesting work. Reviewer VUaS mentions that "the experiments could be strenghened by adding more comparisons". Reviewers also find limited applicability to real-world scenarios. Despite those weaknesses, as it stands, the claims in the paper are accurate and supported by evidence, and the current set of experiments demonstrates the promisingness of the approach. The theoretical contributions in the paper are also sound and well explained.